# Adaptive modulation of brain hemodynamics across stereotyped running episodes

Antoine Bergel [1,2✉], Elodie Tiran[2], Thomas Deffieux[2], Charlie Demené[2], Mickaël Tanter[2,3✉] &
Ivan Cohen [1,3✉]

During locomotion, theta and gamma rhythms are essential to ensure timely communication between brain structures. However, their metabolic cost and contribution to neuroimaging signals remain elusive. To finely characterize neurovascular interactions during locomotion, we simultaneously recorded mesoscale brain hemodynamics using functional ultrasound (fUS) and local field potentials (LFP) in numerous brain structures of freely-running overtrained rats. Locomotion events were reliably followed by a surge in blood flow in a sequence involving the retrosplenial cortex, dorsal thalamus, dentate gyrus and CA regions successively, with delays ranging from 0.8 to 1.6 seconds after peak speed. Conversely, primary motor cortex was suppressed and subsequently recruited during reward uptake. Surprisingly, brain hemodynamics were strongly modulated across trials within the same recording session; cortical blood flow sharply decreased after 10–20 runs, while hippocampal responses strongly and linearly increased, particularly in the CA regions. This effect occurred while running speed and theta activity remained constant and was accompanied by an increase in the power of hippocampal, but not cortical, high-frequency oscillations (100–150 Hz). Our findings reveal distinct vascular subnetworks modulated across fast and slow timescales and suggest strong hemodynamic adaptation, despite the repetition of a stereotyped behavior.

[1] Sorbonne Université, CNRS, INSERM, Institut de Biologie Paris Seine-Neuroscience, 75005 Paris, France. [2] Physique pour la Médecine Paris, INSERM
U1273, ESPCI Paris, CNRS FRE 2031, PSL Université Recherche, Paris, France. [3]These authors contributed equally: Mickaël Tanter, Ivan Cohen.
✉email: antoine.bergel@espci.fr; mickael.tanter@espci.fr; ivan.cohen@upmc.fr

From the early days of electroencephalography (EEG), brain rhythms have been observed in a wide range of models and used as markers to characterize behaviors such as locomotion, sleep states, attention, or cognitive control[1,2]. These neural oscillations support timely communication between distant brain areas by providing windows of opportunity for efficient spike synchrony – a process known as phase synchronization or communication through coherence[3,4] – and their disruption often is a hallmark of pathological conditions like epilepsy, schizophrenia, or Parkinson's disease[5]. Interestingly, many such oscillations are not stationary but instead circulate across brain regions: during locomotion in rodents, theta waves travel along the septotemporal axis of the hippocampus[6,7], slow waves during NREM sleep travel from anterior towards posterior sites[8] and sleep spindles in humans rotate in turn along temporal, parietal, and frontal cortical sites[9]. Exploring these phenomena is thus challenging from an experimental standpoint, due to the limited scope of high-density electrophysiology recordings and the specific difficulty to image global brain activity during natural behavior.

Locomotion is a prime example of natural behavior associated with a typical rhythmic neural pattern: theta rhythm (6–10 Hz). It is observable across brain structures (hippocampus, entorhinal cortex, subiculum, striatum, and thalamus) and species (bats, cats, rabbits, dogs, rodents, monkeys)[10] when an animal engages in walking, running, flying, whisking, and foraging behaviors or enters rapid-eye-movement sleep[11]. Theta oscillations support multiple functions: they are critical for sensorimotor integration[12], contextual information encoding[13], hippocampal–cortical communication[14], structuring the firing of place cells into 'theta sequences'[15], and memory consolidation during REM sleep[16]. Interestingly, theta cycles often contain nested faster (gamma) oscillations in the 30–150 Hz range which exhibit cross-frequency phase-amplitude coupling[17]. In rodents, hippocampal gamma oscillations have been divided into three different subtypes, namely low gamma (30–50 Hz), mid gamma (50–100 Hz), and high gamma or epsilon (100–150 Hz) sometimes also called high-frequency oscillations (HFO) (110–160 Hz). These three oscillatory patterns are generated by different brain structures[18] and likely serve different functions like memory encoding or retrieval[19]. Importantly, all of these gamma oscillations occur during locomotion and high gamma/HFO strongly differ from ripple oscillations – which are observed when an animal is immobile (drinking, grooming, getting ready to move) or sleeps – in terms of amplitude, region of occurrence, activity time-course, and, as mentioned, associated brain state[20].

In addition to electrode studies, neuroimaging methods have been able to record mesoscopic and macroscopic brain activity during head-restrained locomotion, relying on the use of treadmills and virtual navigation. These seminal studies have established that locomotion strongly modulates brain activity in distributed cortical[21], hippocampal[22] and more generally spinal, subcortical, and cortical populations[23], and triggers prominent calcium signals in cortical[21] and cerebellar astrocytes[24,25]. More recently, locomotion has been associated with elevated visual cortex firing rate[26] and remapping of local activity to distal distributed cortical networks[27]. In parallel, intrinsic optical imaging studies have revealed that locomotion modulates frontal and somatosensory cortical areas differently[28], that it is associated with dural vessel constriction[29] and cortex wide increases in brain oxygenation[30]. Unfortunately, these studies fall short in providing large-scale brain activity in deep networks (except in rare cases[23]) due to the poor penetration of optical techniques and to the intrinsic difficulty to adapt locomotion behavior to functional Magnetic Resonance Imaging (fMRI). To date, the best insights into locomotion-related functional activations in the entire brain come from autoradiography[31], miniature portable PET systems[32],

and more recently single-photon emission computed tomography[33]. These studies have revealed considerable activations in posterior cortex, dorsal hippocampus, and striatum, but suffer from low temporal resolution or provide only static images taken at a given point in time.

So, what is the cost of locomotion? And why is such a basic behavior, present in the oldest species and persistent after total decortication[34], accompanied with such intense rhythmic neural activity? It is reasonable to believe that this cost is elevated due to the wealth of processes it triggers in addition to neural and astrocytic ones: coordinated muscular activity, elevated heart and respiration rates, and increased neuromodulator regulation. Also, though theta and gamma rhythms are not essential to locomotion per se, the fact that they correlate with speed and are observable in numerous brain structures, suggests that large-scale cell assemblies are active and coordinated during locomotion. To what extent the activity of these cells is reflected in brain hemodynamics and the relative cost of different brain rhythms which involve different cell types are questions of the utmost importance to interpret blood-oxygen-level-dependent (BOLD) and hemodynamic signals[35]. The common consensus on neurovascular coupling is that local neural activation triggers vasodilation (via decreased vascular tone), resulting in both higher blood velocity and higher blood volume to meet higher energy demands and ensure waste products removal[36]. But the exact mechanisms of this coupling are complex and may involve several pathways in parallel: namely a long-range direct neuronal pathway, and two local ones involving respectively interneurons and astrocytes[37–39]. In addition, neurovascular interactions show regional dependence[40], non-linearity[41], and cell-type specificity[42]. Many studies have focused on neurovascular coupling during anesthesia which abolishes behavior and dramatically alters hemodynamics[43]. This results in an important knowledge gap in the mechanisms of neurovascular interactions during complex behavior in general and during natural locomotion in particular.

To quantify the neurovascular interactions in distributed brain networks (including deep structures) during natural locomotion, we used the emerging functional ultrasound (fUS) imaging modality together with extracellular recordings of local field potentials (LFP) and video monitoring in freely moving rats. fUS can monitor brain hemodynamics over prolonged periods of time in mobile animals, which makes it a well-suited tool for functional imaging of complex behavior such as locomotion[44]. Its key features include a large field of view, high spatial (in-plane: 100 μm × 100 μm, out-of-plane: 400 μm) and temporal (200 ms) resolutions and high sensitivity to transient events. We found that locomotion activates a wide network including dorsal hippocampus, dorsal thalamus, retrosplenial, and parietal cortices in a precise dynamic sequence consistent across recordings. Conversely, primary motor cortex was suppressed during locomotion and activated subsequently during reward uptake. Hippocampal theta and high-gamma rhythms (50–150 Hz) were highly correlated to vascular activity in the hippocampus and thalamus, but only moderately in the neocortex. Intriguingly, brain hemodynamics were strongly modulated across trials and showed a sharp adaptation (reduction in response amplitude) in cortical regions after 10–20 runs concurrent with a more gradual linear potentiation in the hippocampus. Importantly, behavioral parameters like running speed and head acceleration together with hippocampal theta rhythm remained constant between trials and we could relate this vascular reshaping in the hippocampus to an increase in the power of hippocampal high-frequency oscillations within individual recordings. Taken together, these results provide new insights into the understanding of neurovascular coupling during locomotion and reveal the complexity and richness of vascular dynamics despite stereotyped behavior.

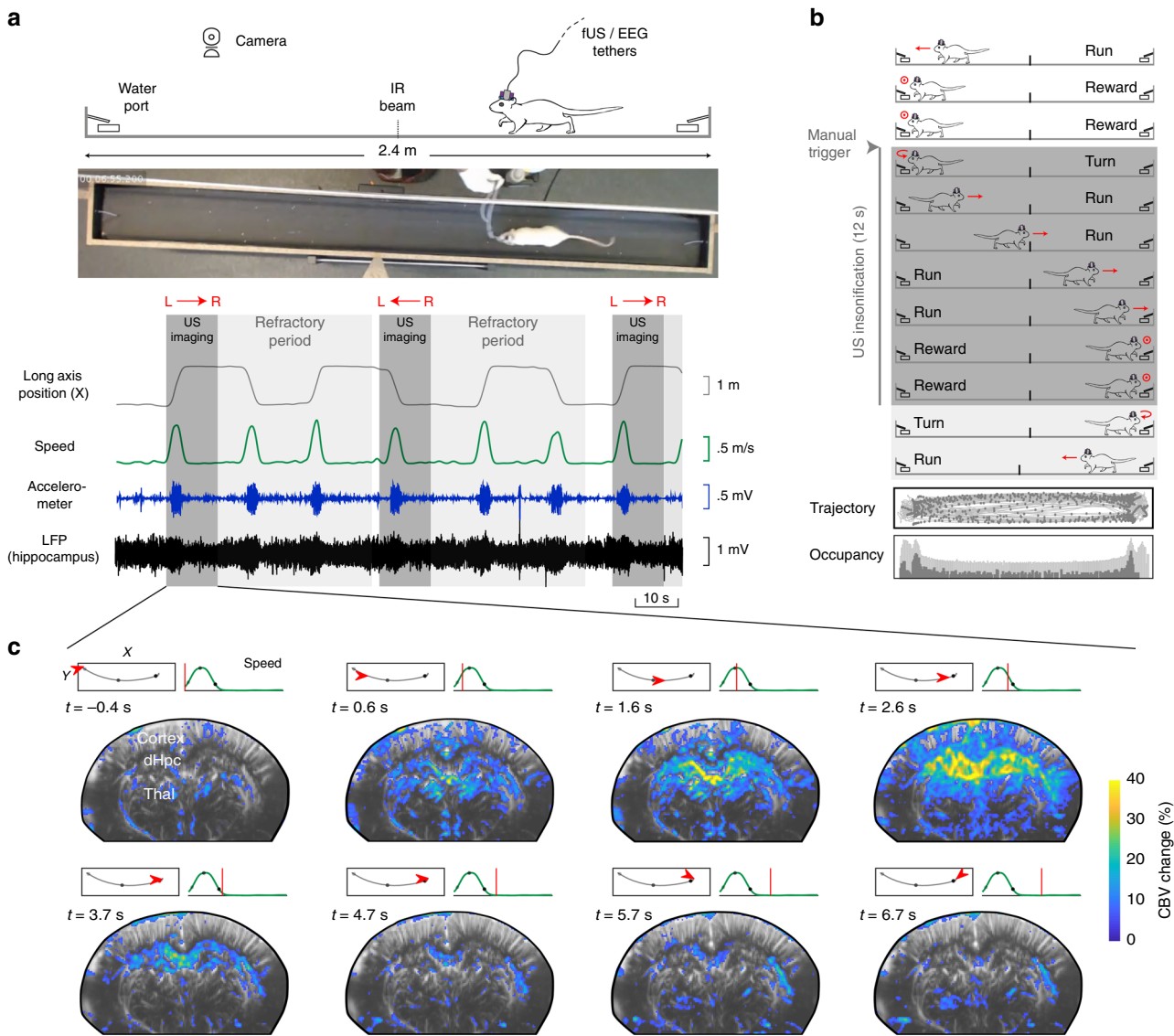

**Fig. 1 Simultaneous functional ultrasound/LFP/video recordings reveal single-run hemodynamics in deep brain structures. a** Schematic and picture of the fUS–LFP–video setup. Rats are trained daily to run back and forth on a 2.4 m linear track for water reward, given at both ends. Video, electrophysiology, and accelerometer data are acquired continuously. Ultrasound data is acquired in bouts lasting 12 s (shaded dark gray), interleaved with a 40-s refractory period (shaded light gray), during which ultrasound data is beamformed. Animal position on the track (gray) and speed (dark green) are extracted offline concurrently with LFP processing. **b** Details of single-run imaging. Ultrasound acquisition is triggered manually by the experimenter, when the animal initiates its turn. A typical 12-s imaging sequence is composed of Turn (1 s) – Run (3–4 s) – Reward (7–8 s). A typical recording session lasts 30–40 min and contains 80–100 trials, a third of which are imaged (2/3 occurring during the refractory period). Trajectory and occupancy of the track are displayed at the bottom for the full session (light gray) and for the imaged runs (dark gray: one Doppler frame, 200 ms compounding). **c** Spatiotemporal dynamics of the vascular network during a single run. A typical coronal plane (Bregma −4.0 mm) reveals cerebral blood volume (CBV) in cortical, hippocampal, and thalamic regions. For each 12-s US insonification, we formed 60 Power Doppler frames (8 frames showed, 5 Hz total sampling) based on ultrasound echoes. This reveals prominent activation in the dorsal hippocampus (30–40 % CBV change), peaking 2.0–2.5 s after fUS onset, that is 1.5–2.0 s after run onset. Timing from run onset is given for each frame. Top left: Animal position on the track (red arrow) overlaid on trajectory (gray line) Top right: Animal speed as a function of time (red line: current position). Gray dots mark run onset (light gray), run peak (gray), and run end (dark gray).

## Results

**Voluntary locomotion activates a brain-wide vascular network.** To reveal the vascular networks recruited during natural locomotion and finely characterize the neurovascular interactions between hippocampal rhythms and brain hemodynamics, we used fUS in mobile rats combined with electrophysiological recordings of local field potentials (LFP) in the dorsal hippocampus and video (Fig. 1a), using previously introduced fUS–LFP–video setup[44]. Electrode bundles were implanted in seven animals in posterior sites unilaterally (retrosplenial cortex,

dorsal hippocampus, dorsal thalamus) and two animals were also implanted in contralateral anterior sites (primary motor cortex) (Supplementary Fig. 1). Depending on the setup and apparatus used, fUS imaging can probe cerebral blood volume (CBV) and cerebral blood flow (CBF) over multiple brain regions, including deep structures, over a single plane or a full volume[45,46] with a spatial resolution depending on the frequency of the ultrasonic probe (15 MHz linear probe: 100 x 100 x 400 microns resolution) and a temporal resolution up to 500 Hz, when Doppler frames are formed using sliding windows from compound images. In this

experiment, we used a 'burst sequence' that alternates 12-s insonification periods during which ultrasound echoes are sent and received at 500 Hz until memory saturates, with refractory periods for ultrasound images to be 'beamformed' and transferred, during which vascular activity cannot be monitored. The onset of ultrasound acquisition was triggered manually by the experimenter, when the animal initiated a body rotation to start a new run in the opposite direction on the linear track (Fig. 1b). Such a setup enabled the monitoring of CBV variation during single runs at $100 \times 100 \times 400\ \mu m$ spatial and 200 ms temporal resolution in multiple brain structures over a single imaging plane. On average, 2 out of 3 runs occurred during the refractory period and could not be imaged, leading to an average of 30–35 trials over 100 runs per recording, with a global balance between running directions. The tracking of the animal's position enabled to realign all 12-s fUS recording periods towards a common time origin, defined as the onset of each run (Fig. 1c). A single trial was defined as a run that was correctly captured from onset to end in the 12-s sequence, with run onset and end defined as a 10% threshold of peak speed (see Methods).

To reveal the hemodynamic responses to locomotion in a large number of regions while keeping sufficient statistical power, we chose to perform repeated ultrasound acquisitions in different animals over three typical recording planes: two coronal planes and a diagonal one. In a first set of experiments, we imaged posterior brain structures over a coronal section (AP = −4.0 mm, $n = 11$ rats, 25 recordings) that intersected the dorsal hippocampus, thalamus, hypothalamus, and cortex, including auditory (AC), primary somatosensory barrel field (S1BF) lateral parietal association (LPtA) and retrosplenial (RS) cortices, and a diagonal section tilted 45° relative to coronal view ($n = 4$ rats, 22 recordings) that intersected the hippocampus in full (dorsal, intermediate, and ventral), thalamus (dorsal and ventral), cortex (anterior, midline, and posterior) and caudate Putamen (CPu) (Fig. 2a Left Lane – Supplementary Fig. 2). In a second set of experiments, we imaged anterior structures in the brain over an anterior coronal section (AP = + 3.0 mm, $n = 3$ rats, 15 recordings) that intersected the primary (M1) and secondary (M2) motor cortices, anterior cingulate (Cg1), prelimbic (PrL), and infralimbic (IL) cortices and CPu (Fig. 2a Right Lane). We then registered two-dimensional vascular planes over two reference atlases (Paxinos atlas for coronal planes; Waxholm MRI atlas for diagonal planes) to derive regions of interests (ROIs) as described previously[47] and spatially averaged the CBV signal in these ROIs. We re-aligned all trials for all recordings to the onset of each run and computed temporal averages of brain hemodynamics of all aligned trials and compared them with behavioral parameters such as running speed and head acceleration. This revealed prominent CBV increases that were time-locked to run onset in the dorsal hippocampus bilaterally though both inter-trial and inter-individual variability were present. Activations in the cortex and thalamus were of low amplitude due to the spatial averaging in such large regions (Fig. 2a Left Lane). In the primary motor cortex, we observed the inverse pattern: CBV signal showed a prominent reduction during locomotion and a subsequent activation right after run end, when the animal licks to get the water reward (Fig. 2a Right Lane). This is in line with previous studies that reported a decrease in motor cortex activity during automated behavior[48] and a positive correlation between motor cortex recruitment and task engagement[49]. To finely characterize the vascular network activated during locomotion we computed the average hemodynamic responses for 16 sub-regions identifiable on the posterior coronal section (11 animals, 22 recordings, 384 trials) and for four sub-regions identifiable on the anterior coronal section (3 animals, 12 recordings, 928 trials), together with the average running speed. This revealed prominent

activations in lateral parietal and retrosplenial cortices, all dorsal hippocampal sub-regions with strongest activation found in the dentate gyrus, and dorsal thalamus. Interestingly, the primary motor cortex (and secondary motor cortex, though to a lesser extent) was negatively modulated by locomotion whereas other frontal cortices and caudate putamen did not show prominent modulation (Fig. 2b). A similar analysis over diagonal plane recordings revealed that parietal cortical regions and posterior cortical regions were also active during locomotion, that dorsal hippocampus was strongly recruited while ventral hippocampal activations were absent, and that such dissociation was also visible in the thalamus (Supplementary Fig. 2). Hemodynamic responses were notably symmetrical and showed no obvious dependence on running direction (Supplementary Fig. 3).

**Distant brain regions are activated in a dynamic sequence with delays ranging from 0.8 to 1.6 s after peak running speed.** In order to quantify the coupling between brain hemodynamics and running speed and to reconstruct the precise sequence of activation during voluntary locomotion, we computed cross-correlation functions between CBV pixel signals and running speed (Fig. 3a). Importantly, cross-correlation analysis is not precluded by the discontinuity of CBV signals (due to the refractory period imposed by the burst sequence), because speed was monitored continuously meaning that CBV–speed correlations can be computed for all time delays, in a similar fashion as what would be done if CBV signal were continuous. In order to precisely assess the delays of speed–CBV correlations, we reconstructed high-definition Doppler movies using sliding 200-ms sliding windows, with 190-ms overlap, leading to one Doppler frame every 10 ms. Speed–CBV correlation maps were computed for all correlation lags between −1.0 s to 5.0 s with a step size of 10 ms, leading to a precise temporal reconstruction of the sequence activated during locomotion over the two typical coronal and diagonal recordings (Supplementary Movie 1–3). For all pixels in a given recording, we extracted the coordinates of the peak of the cross-correlation function leading to a maximal correlation map $R_{max}$ and a corresponding delay map $T_{max}$ (Fig. 3b). Strong cross-correlation values were prominent in the dorsal hippocampus, dorsal thalamus, and in parietal and posterior cortices of the maximal correlation map. This approach led to a precise time sequence of individual pixel responses to locomotion, but did not allow for statistical comparison across recordings, due to pixel mismatch resulting from heterogenous probe position across sessions. To circumvent this, we used the same approach for regional hemodynamics.

In coronal observations, both individual (Fig. 3c Left Lane) and group distributions (Fig. 3d Left Lane) showed that dorsal thalamus (1.01 s ± 0.10 s) and retrosplenial cortex (1.16 s ± 0.15 s) peaked earlier than hippocampal regions. We also found that dentate gyrus peaked earlier (1.42 s ± 0.10 s) than CA3 (1.58 s ± 0.13 s) and CA1 (1.63 s ± 0.10 s) regions. The delay between dentate gyrus and dorsal thalamus was found significant (Wilcoxon signed-rank test, $P < 0.001$) as well as the delay between DG/CA3 ($P < 0.05$) and DG/CA1 ($P < 0.01$). This pattern was bilateral and robust across recording sessions. Qualitatively, dentate gyrus response typically started earlier and lasted longer than CA1/CA3 responses, which were shorter, but of higher amplitude. This suggests that hippocampal subfields are perfused by at least two distinct vascular networks. Consistent with earlier observations, ventral thalamus displayed moderate anti-correlation relative to animal speed[44]. This dynamic sequence of vascular activation was also visible over diagonal planes (Fig. 3c, d Right Lane) starting in the retrosplenial cortex (posterior: 0.85 s ± 0.27 s, anterior: 0.91 s ± 0.27 s) and in the

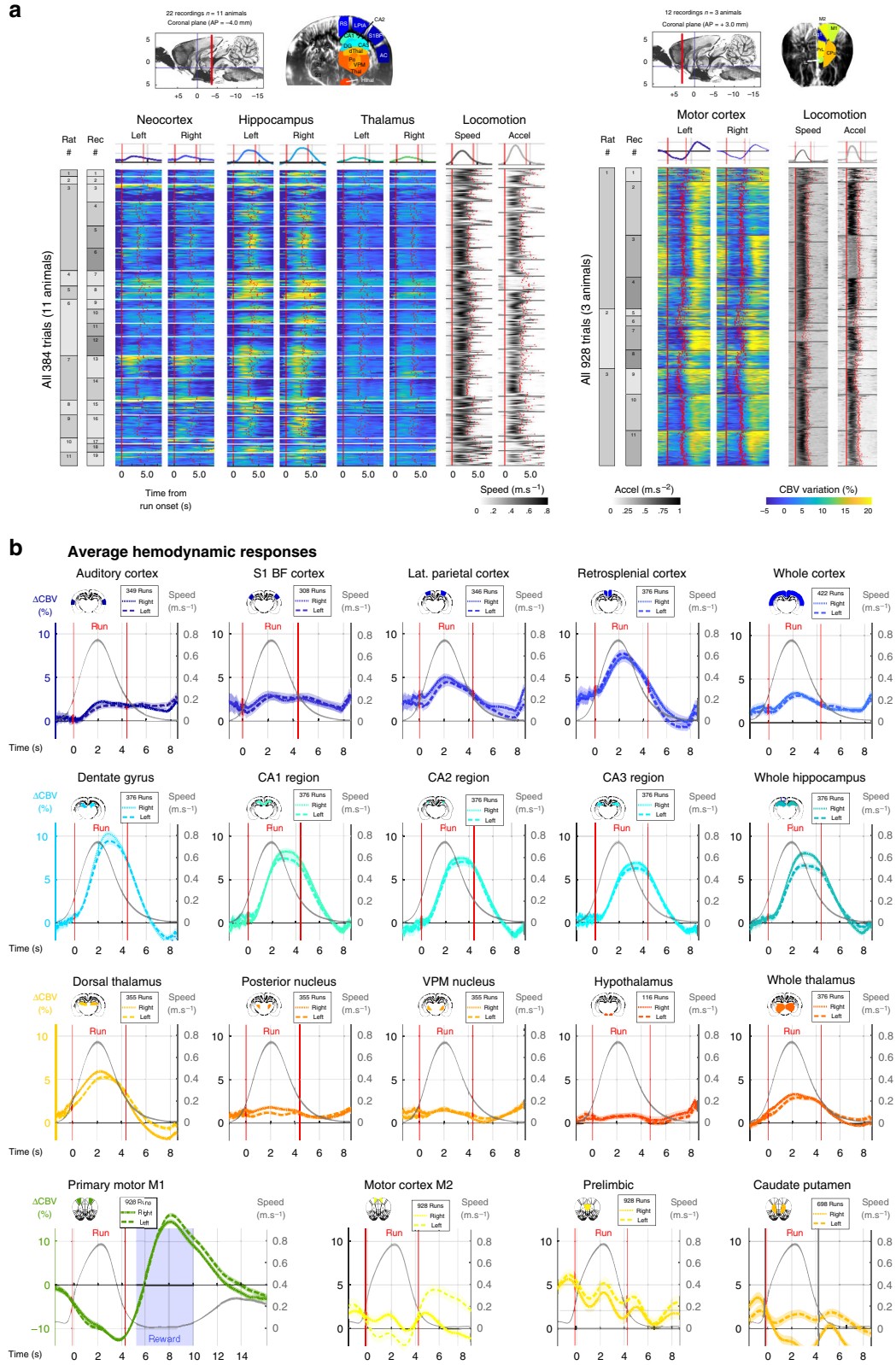

**a**

**b** Average hemodynamic responses

dorsal thalamus (0.93 s ± 0.15 s), then reaching dentate gyrus (1.15 s ± 0.06 s) and finally CA3 (1.22 s ± 0.08 s) and CA1 regions (1.38 s ± 0.08 s). This time, only the delay between dentate gyrus and dorsal thalamus was found significant (Wilcoxon signed-rank test, $P < 0.05$) as well as the delay between DG and CA1 ($P < 0.05$). Interestingly, relative delays across regions were consistent across coronal and diagonal planes, but absolute speed–CBV delays tended to be shorter in diagonal planes. This can be explained by the fact that atlas-registration is more prone to errors in the diagonal recordings, especially in hippocampal subfields but also by the fact that large sub-regions (like DG and CA1) might not activate homogenously over the whole diagonal

**Fig. 2 Unrestrained locomotion activates or suppresses a vast network of brain regions. a** Bulk representation of single-trial hemodynamic responses to locomotion along two typical coronal recording planes (Left: Plane 1, Bregma −4.0 mm, right: Plane 2, Bregma +3.0 mm) acquired from two separate groups of animals. For each plane, a typical Power Doppler image is shown with the corresponding regions of interests, derived from atlas registration. (Left) Total of 384 trials (19 recordings, 11 animals) revealing hemodynamic responses in posterior cortices, dorsal hippocampus, thalamus, and hypothalamus. The CBV signal is a spatial average over the corresponding region of interest and is expressed in % of change relative to baseline. For each run, the onset of movement is used as a temporal reference (zero-timing) and all trials are aligned to run onset (see Methods). We can thus derive an average hemodynamic response for all trials (top). The same approach is performed for locomotion parameters: running speed (left) and head acceleration (right). Hemodynamic responses are symmetrical and display strong activations in cortical and hippocampal regions. (Right) Total of 928 trials (11 recordings, 3 animals) revealing hemodynamic responses in anterior cortices and striatum. Interestingly, primary motor cortex was strongly suppressed during locomotion and subsequently activated during reward uptake (water drop). Other frontal cortices and striatum do not display significant activation during locomotion. MRI plate reprinted from ref. [88], Copyright (2014), with permission from Elsevier. **b** Average hemodynamics responses over multiple sub-regions. Using the same approach as in **a**, we computed average responses to locomotion in 16 sub-regions and three groups of regions. We overlay running speed (gray) to visualize both the response intensity and temporal delays to locomotion. Overall, strong bilateral activations are found in all dorsal hippocampus sub-regions (stronger and earlier in the dentate gyrus), retrosplenial cortex, and dorsal thalamus. The primary motor cortex shows both suppression during locomotion and activation during reward, probably to the different types of motor behavior required in both actions. For each region the number of runs can differ slightly as all 24 sub-regions were not always visible on each recording. (Plane 1) RS retrosplenial cortex, LPtA lateral parietal association cortex, S1BF primary somatosensory barrel field cortex, AC auditory cortex, DG dentate gyrus, CA1–CA2–CA3 region, dThal dorsal thalamus, Po posterior thalamic nucleus, VPM ventroposterior thalamic nucleus, Thal thalamus, Hthal hypothalamus. (Plane 2) M1 primary motor cortex, M2 secondary motor cortex, Cg1 anterior cingulate cortex, PrL prelimbic cortex, IL infralimbic cortex, CPu caudate putamen. Error bands correspond to the mean values ±sem. Source data are provided as a Source Data file.

plane. Overall, the precise dynamic sequence was remarkably conserved across coronal and diagonal group recordings. Our results show a pattern of activation in the dorsal two-thirds of the hippocampus. Ultimately, we reconstructed a post-hoc reference activation sequence based on the average measures ($R_{max}$,$T_{max}$) for each ROI from all individuals (Fig. 3c, d Bottom line) to give an overview of the spatiotemporal dynamics of the sequence associated with running speed over the two chosen recordings planes. The same analysis was performed for other seed regressors, such as head acceleration and LFP power envelopes signals (Supplementary Fig. 4).

**Neurovascular interactions are complex and region-dependent during natural locomotion.** fUS enables the study of neurovascular coupling in several distant brain regions simultaneously, which is usually confined to electrophysiology–fMRI studies at rare exceptions. We leveraged this to investigate the nature of local and distant neurovascular interactions between hippocampal and cortical rhythms and vascular activity: we thus computed LFP–CBV cross-correlation functions between regional signals (neocortex, dorsal hippocampus, and thalamus for posterior recordings; primary, secondary motor cortices and caudate putamen for frontal recordings) and LFP signal filtered in four typical LFP bands: theta (6–10 Hz), low gamma (20–50 Hz), mid gamma (50–100 Hz), and high gamma (100–150 Hz) both at the hippocampal and motor cortex recording sites (Fig. 4a). To smooth the rapid dynamics of LFP data, we computed the power envelope and convolved it to a Gaussian kernel (see Methods). Hippocampal theta, mid and high gamma bursts of activity were clearly visible on the spectrogram when the animal prepared to or actually engaged in running on the linear track, but not during reward uptake (Supplementary Fig. 5). Theta and gamma showed phase-amplitude cross-frequency coupling, with low-gamma peaking on the ascending phase of theta, mid-gamma power being maximal at theta peak and high-gamma maximal at theta trough, which is in accordance with previous results in the literature[18,50]. In posterior regions, hippocampal theta activity was more strongly associated with vascular responses in the hippocampus and thalamus ($R_{max} = 0.51$, $R_{max} = 0.40$), and markedly less in the whole cortex ($R_{max} = 0.28$). Consistent with previous observations during REM sleep, we found that hippocampal low gamma did not correlate with CBV in any brain region[47], while mid- and high-gamma bands showed moderate

correlation with CBV responses only in the hippocampus ($R_{max} = 0.41$ in both regions).

As expected, hemodynamics in frontal regions were modulated by hippocampal rhythms (consistent with their modulation by running speed) and primary motor cortex displayed a strong decorrelation to hippocampal theta activity for delays in the 1–2 s window ($R_{min} = 0.45$) (Fig. 4b). This coupling was present but less prominent for mid and fast gamma oscillations which also correlate with speed. Other frontal regions like M2 and striatum showed a positive coupling with hippocampal theta and fast gamma. This inversion of neurovascular profile was extremely sharp and could be used to delineate subregions extremely clearly (Supplementary Movie 3). Moreover, motor cortex activity did correlate positively to hippocampal rhythms for negative delays ($T_{max} = -1.5$ s) or longer ones (> 5 s) which is interesting: it could simply be due to the rhythmicity of the task, but the fact that we found stronger correlations for negative delays than for longer ones suggest that M1 hemodynamics might anticipate LFP hippocampal activity. Last, we could observe layer-specific propagation in the motor cortex showing that vascular signals in the superficial and deep cortical layers were not exactly in phase, but the intrinsic rhythmicity of the task impeded further analysis.

Local neurovascular interactions in the primary motor cortex confirmed previous results in the literature showing that CBV signal is decoupled from LFP activity during locomotion[28]: we found a decoupling from CBV signal with all LFP signals, with a strongest effect for mid and fast gamma oscillations (Fig. 4c). Indeed, theta was not prominent but peaks in mid gamma (50–100 Hz) power were found during locomotion and these oscillations displayed phase-amplitude coupling relative to theta (Supplementary Fig. 5, Supplementary Movie 6). Again, other frontal regions displayed a positive coupling to cortical LFP signal, strengthening the singularity of neurovascular interactions in the primary cortex during locomotion. Taken together, these results show that LFP and CBV signals are not unequivocally related to one another (hence LFP cannot be easily inferred from CBV data and conversely) and suggest that the primary motor cortex possesses specific mechanisms to display such a peculiar profile in terms of neurovascular interactions.

**Locomotion-related hemodynamics are strongly modulated despite stereotyped running.** Because of its high sensitivity and

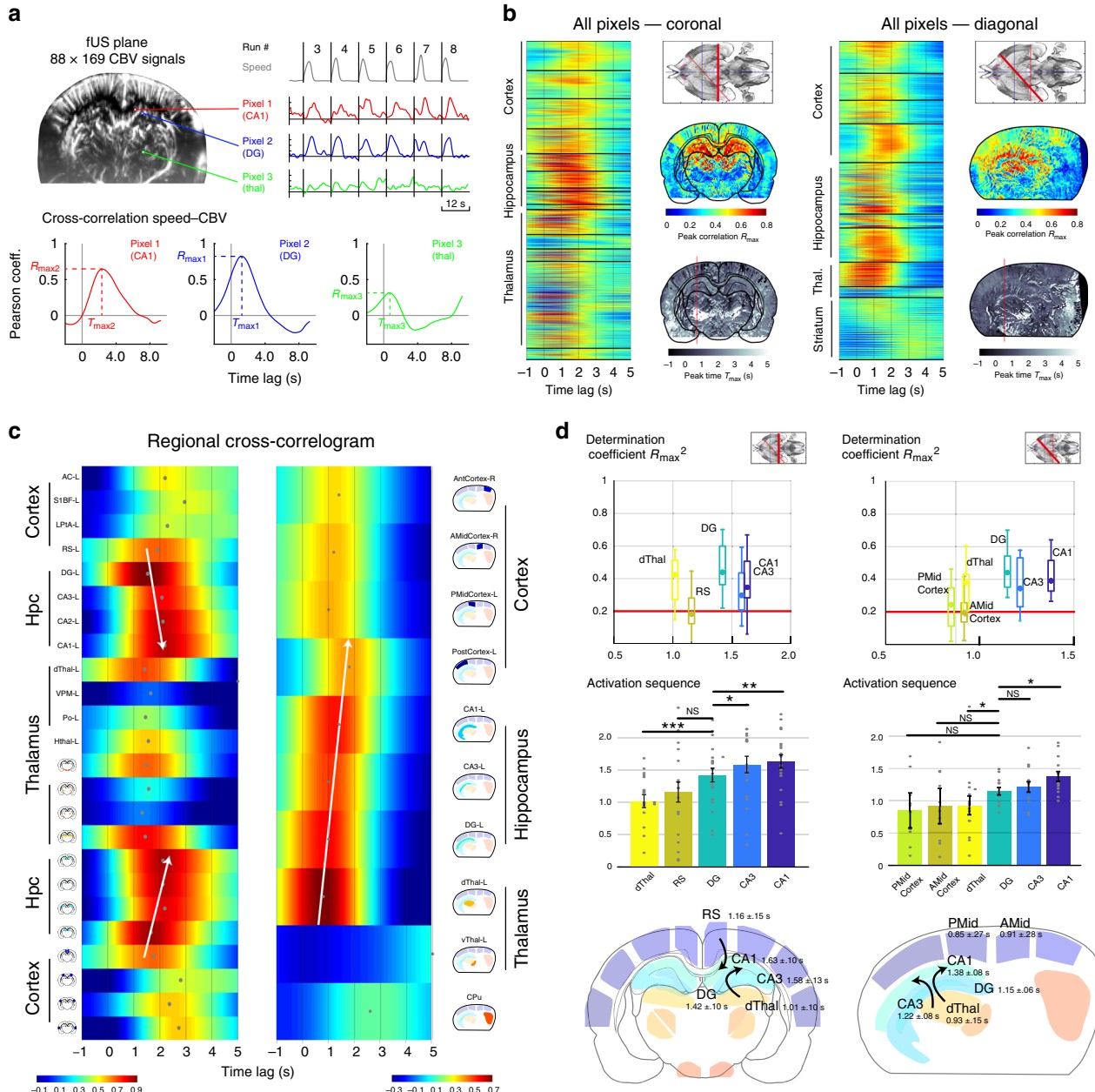

high temporal resolution, transient events can be monitored with fUS circumventing the need for temporal averaging routinely implemented in BOLD-fMRI or PET studies. We thus interrogated whether brain hemodynamics were invariant or modulated across trials within a single recording session and, if such effect was present, whether it affected brain regions to the same extent. We repeated the approach presented in Fig. 2 that consisted in re-aligning all trials onto a common time origin taken as the onset of each run, but this time we focused on each individual recording independently. For each recording, we first sorted all trials in three different time groups, respectively, including all early trials (0–15 min from session onset), intermediate trials (15–30 min) and late trials (>30 min) in order to compare brain activity across regions and time. Individual pixel-based peak correlation maps showed an increased speed–CBV coupling in the dorsal hippocampus in the intermediate and late groups versus the early time group, concurrent with a decreased speed–CBV coupling in the cortex and in the dorsal thalamus,

though the first was more gradual than the second one (Fig. 5a). This effect was consistent across recordings: the average hemodynamic responses for all coronal recordings showed a strong amplitude increase in intermediate/late time groups compared to the early one both in the thalamus and hippocampus, a moderate amplitude reduction in the cortex, while the hypothalamic response remained constant across time groups (Fig. 5b). Notably, this strong modulation across individual trials within the same recording session occurred while behavioral parameters such as running speed or head acceleration remained constant. This effect was very prominent and clearly visible both at the scale of individual runs (Supplementary Fig. 6) and in the bulk representations of single-trial responses across regions (Supplementary Fig. 7). We also displayed Doppler movies from five consecutive runs taken during the early, intermediate, and late period of the recording sessions for two individuals in the posterior coronal plane (Supplementary Movie 4–5) and one individual for the anterior coronal plane (Supplementary Movie 6).

**Fig. 3 Correlation analyses reveal a precise and dynamic activation sequence along the dorsal thalamus, retrosplenial cortex, and hippocampal subfields. a** Brain hemodynamics are monitored along coronal (Bregma = −4.0 mm, $n = 7$ animals, 22 recordings) and diagonal sections (Delta = 45–60°, $n = 4$ animals, 20 recordings,). A typical recording plane contains 15,000 voxels from which a CBV signal can be derived and synchronized with speed (Top right, example of 3 pixels, red: CA1 region, blue: Dentate gyrus, green: Po nucleus). For each pixel, we compute the cross-correlation function (time-lags from −1.0 s to 5.0 s) between speed and CBV signal. Then we extract the coordinates of the peak, leading to a measure of the peak correlation $R_{max}$ (y-coordinate) and corresponding peak time $T_{max}$ (x-coordinate). **b** The approach illustrated in **a** is performed for all pixels over the coronal (left) and diagonal (right) recordings. After atlas registration, we can re-order the cross-correlations by regions. Strong correlations are clearly visible is some cortical regions (RS) and thalamic (dorsal) regions, while all dorsal hippocampal regions strongly correlate with speed. Note the different delays across brain regions. For each recording, we can extract a peak correlation map (top) and a peak timing map (bottom). Note the consistency between the two recording planes taken from two different animals. MRI plate reprinted from ref. [88], Copyright (2014), with permission from Elsevier. **c** Regional correlograms for the two recordings shown in **b**. We derived an average CBV signal across each region and computed its cross-correlation with speed, similarly to **b**. This reveals a dynamic pattern of activation at the sub-second timescale. Note the clear propagation along the regions of the dorsal hippocampus bilaterally (white arrows). This pattern is also visible on the diagonal plane. Ventral thalamus, striatum, and hypothalamus respond more moderately to locomotion. **d** Group Distributions of the parameters shown in **c** for all coronal (Left, 25 recordings, 11 animals) and diagonal recordings (Right, 22 recordings, 7 animals). (Top) Synthesis of determination coefficients ($R_{max}^2$) ordered by time relative to peak speed. For the sake of clarity, we only show the strongly responsive regions (criterion $R_{max}^2 > 0.2$). Whisker plots show the median (center), 25th and 75th percentiles (boxes) and 1st and 99th percentile (whiskers). (Middle) Group distributions of the peak time $T_{max}$ ordered in ascending order relative to peak speed. Note the clear sequential propagation (dThal/RS > Dentate Gyrus > CA regions), which is conserved in both coronal and diagonal groups, though the diagonal section timings are slightly shorter. Error bars correspond to the mean values ±sem. (Bottom) Post-hoc reconstruction of the locomotion-related vascular propagation based on the mean peak correlation and peak time for all recordings. Note the consistent delay between dorsal thalamus/dentate gyrus ($P = 0.0002 < 0.001$ $P = 0.0282 < 0.05$, two-tailed signed-rank test, Bonferroni correction) and dentate gyrus/CA1 region ($P = 0.0184 < 0.01$ $P = 0.0482 < 0.05$). Though the absolute timing may vary, the sequence is remarkably preserved across section groups. The list of exact p-values for the different pairs tested for the coronal and diagonal groups are given below: (Coronal) DG/dThal: 0.0002, DG/RS: 0.3996, DG/CA3: 0.0238, DG/CA1: 0.0184. (Diagonal) DG/PMid: 0.1267, DG/Amid: 0.1210, DG/dThal: 0.0282, DG/CA3: 0.6444, DG/CA1: 0.0482. Source data are provided as a Source Data file.

Hippocampal and thalamic hemodynamics on the one hand and cortical hemodynamics on the other, were strongly reshaped across trials but followed opposite patterns. The fact that some regions (ventral thalamus, striatum) showed weak to absent modulation rules out any global effect affecting all regions simultaneously but advocates instead for a strong region-dependent tightly regulated mechanism yet to be defined.

To assess the precise temporal dynamics of this vascular reshaping across brain regions, we divided recordings into 1-min duration bins and computed the mean value of regional CBV signals for all individual trials (in a 200-ms window centered on the peak of activation). We then derived a CBV distribution per bin for all regions across recordings (whisker plots). We observed a strong linear progression in CA1 and CA3 regions between peak amplitude response and session onset (Fig. 6a Top Rows). Comparatively, dentate gyrus showed a slightly lower increase, because the peak response in early trials was already high. In comparison retrosplenial cortex and somatosensory cortex vascular responses were consistently larger in the very early trials (1–5 min) than for the rest of the task. Dorsal thalamus also displayed a linear increase in peak response that reached a plateau after 10 min. Ventral thalamic regions showed relative low amplitude modulation but a gradual increase across trials. We also computed CBV value at trial start and demonstrated that hemodynamic reshaping mainly affects the late component of the hemodynamic response to single trials (Supplementary Fig. 8). In frontal cortical regions, the sharp decrease in the early trials was extremely salient, in particular in secondary motor, anterior cingulate, infralimbic, and prelimbic cortices (Fig. 6a Bottom Row). This was also observable in the primary motor cortex, but the reduction in amplitude was less marked and more linear, whereas caudate putamen was not modulated by trial timing. We computed deviation from the distribution of the first minute and displayed the results of statistical testing on each graph (Mann–Whitney test, $n = 7$ rats, 15 recordings). Overall, this supports the idea of a gradual disengagement of top–down cortical sites (while primary motor cortex was constant) to the benefit of hippocampal and limbic structures as the locomotion task progresses and the animal engages in repetitive behavior.

Linear regression of these histograms allowed us to separate brain regions in a two-dimensional space which revealed three main clusters according to their modulation profile (Fig. 6b). Cluster 1 contained cortical regions that display strong early responses, high variance, and depression across trials. Cluster 2 includes regions like the hippocampus or dorsal thalamus with moderate responses in early trials, strong linear potentiation, and low variance. Cluster 3 groups regions with low to absent modulation (no dependence on trial timing). Interestingly, M1 did not belong to any of these clusters. In individual recordings, we could then perform linear regression for all pixels and segregate them according to the sign of the potentiation slope. In some recordings, vascular depression was observable in the dentate gyrus and cortical sites, while potentiation was present in the CA1/CA3 regions of the hippocampus (Fig. 6c). This shows that vascular reshaping affects differentially hippocampal subfields at least in some individuals. Thus, the function of hemodynamic modulation must to some extent relate to functional or anatomical differences between hippocampal subfields.

**Hippocampal high-frequency oscillations but not theta rhythm nor behavioral parameters account for hemodynamic reshaping.** Up to this point, we demonstrated that brain hemodynamics were strongly reshaped across individual trials within a single recording session, quantified the precise dynamics of this modulation. We then questioned whether vascular reshaping could be accounted for by behavioral parameters or electrophysiological activity, by displaying in parallel vascular, behavioral, and electrophysiological signal in each recording. As shown before, hemodynamics displayed opposite modulation patterns in the retrosplenial cortex, CA1 region, and dentate gyrus while speed and head acceleration remained constant (or slightly decreased). Interestingly, we found that while hippocampal theta activity and low gamma activity remained constant across trials, mid gamma (50–100 Hz) and high frequency oscillations (100–150 Hz) were also positively modulated (Fig. 7a). To provide a quantitative measure of this effect, we computed a median value in each trial for all these variables. This led to column vector of median values,

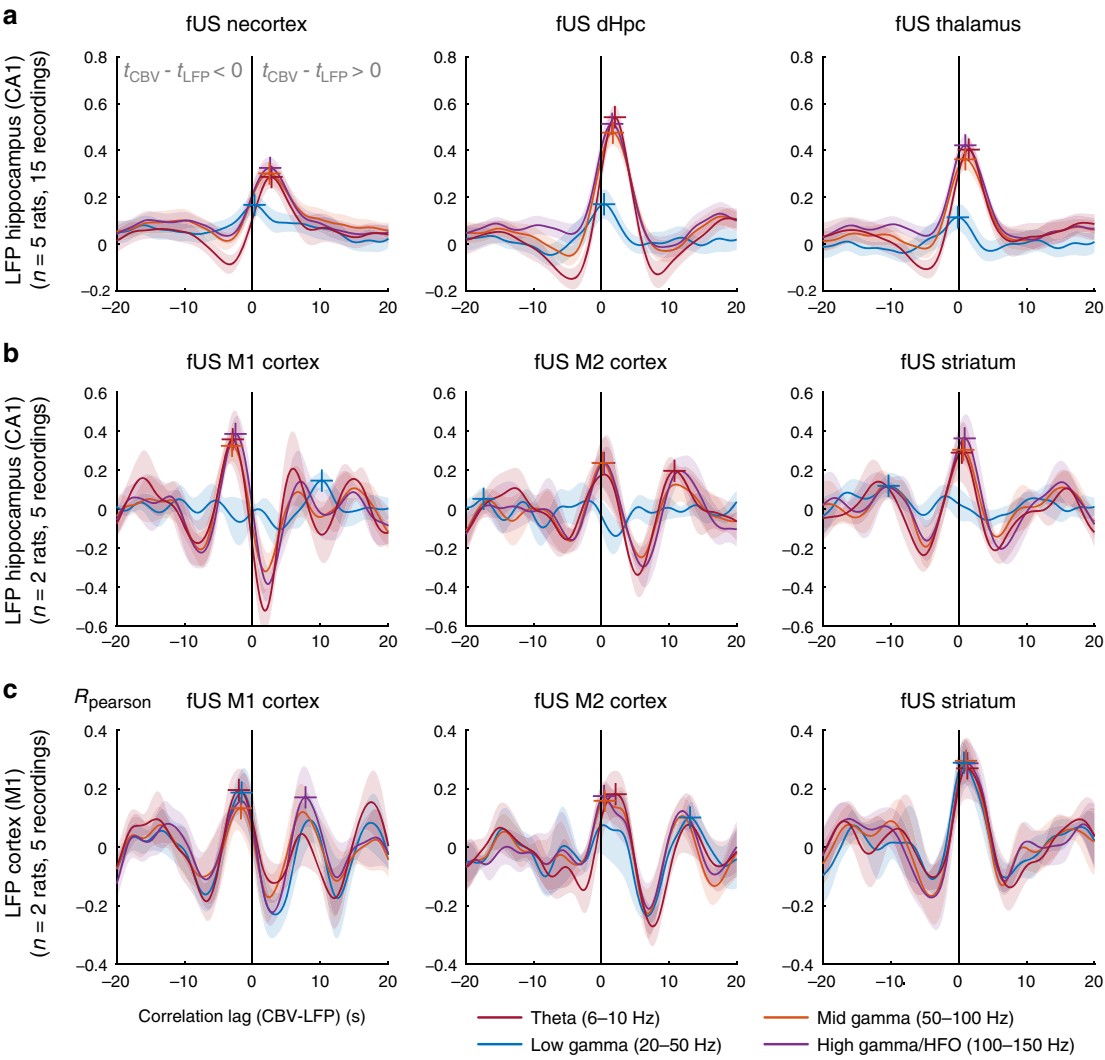

**Fig. 4 Complex relationship between local field potential recordings and regional CBV responses during unrestrained locomotion.** Cross-correlation functions between LFP signal and CBV regional averages for different imaging planes and different electrode locations in the dorsal hippocampus (Top–Middle) and primary motor cortex (Bottom) Cross-correlation functions are computed for all LFP–CBV pairs between the four LFP envelope signals in the theta (6–10 Hz, red) low gamma (20–50 Hz, blue) mid gamma (50–100 Hz, orange), and high gamma bands (100–150 Hz, purple) and three major regions: whole cortex, dorsal hippocampus, and thalamus for posterior recordings (Top) and M1 cortex, M2 cortex, and striatum for anterior recordings (Middle–Bottom). The x-coordinate of the maximum of this function (color crosses) gives the delay between LFP and CBV signals, while the y-coordinate gives the strength of the coupling. **a** In posterior regions, as noted previously, hippocampal low-gamma shows no coupling with CBV signals in either region, both mid- and high-gamma bands show moderate coupling, and theta band shows strong robust coupling especially in the dorsal hippocampus. **b** In anterior brain regions, primary motor cortex shows strong decoupling with distant hippocampal LFP signals in the theta, mid gamma, and fast gamma bands, which is consistent with the inhibition during locomotion. Conversely, both M2 cortex and striatum show positive coupling with hippocampal LFP signals. **c** CBV signal in the primary motor cortex showed decoupling with local LFP signals, in particular for gamma band signals, whereas both M2 cortex and striatum were positively correlated with local LFP signals. Error bands correspond to the mean values ±sem for **a**, **b**, and **c**. Source data are provided as a Source Data file.

the size of which was the number of trials. We then computed correlations between all possible behavior/CBV (4) and electrophysiology/CBV (6 for each electrode) pairs to see whether inter-trial variability (captured in the values of the CBV vectors) did correlate with the variability in behavioral or electrophysiological parameters. We found that trial-to-trial variability was largely independent of behavioral parameters such as running speed or head acceleration in all three directions (Fig. 7b). Low frequency rhythms (theta and low gamma) did not correlate with vascular variability in any brain region. However mid-gamma power correlated moderately with CBV signals in cortical regions, while, interestingly, high gamma (100–150 Hz) and high-frequency oscillations (>150 Hz) correlated strongly with vascular signals in

all dorsal hippocampus subfields and moderately in thalamic and cortical regions. Importantly, this was not the case for electrodes located in the retrosplenial cortex (Fig. 7c), meaning that electrophysiological activity captured in middle and high-frequency oscillations, in the hippocampus specifically, might explain the potentiation patterns observed in vascular signals, while neither behavioral parameters, cortical LFP, or low-frequency hippocampal rhythms could. We computed the statistical significance of the deviation between the cortical and hippocampal correlation distributions and displayed the corresponding p-value for each CBV–LFP pair (Mann–Whitney test). Investigating recording sessions individually confirmed this assertion and we found strong correlations between high-frequency activity and vascular

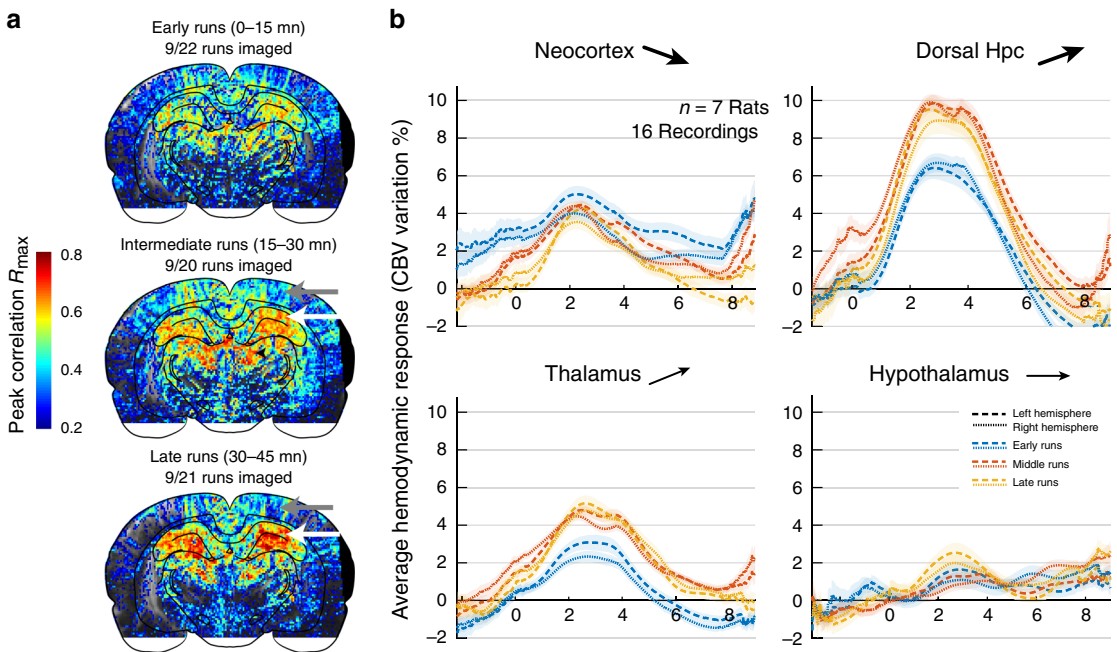

**Fig. 5 Strong region-dependent modulation of hemodynamics across trials within the same recording session. a** Maximal correlation maps relative to animal speed, as shown in Fig. 3b for three groups of trials, namely early (0–15 min from recording onset), intermediate (15–30 min), and late (>30 min) runs. Note the gradual involvement and the strong potentiation in the dorsal hippocampus (white arrows) and the inverse pattern in the cortex (gray arrows). **b** Average hemodynamic responses in four major brain regions grouped by early (0–15 min), intermediate (15–30 min), and late (>30 min) groups. These hemodynamic responses were computed for all coronal recordings that contained at least 24 imaged runs in total (AP = −4.0 mm, n = 7 animals, 16 out of 25 recordings). Neocortex and hippocampus/thalamus regions display opposing modulation patterns, showing that this phenomenon is not a global effect of temperature or reward during the task. Arrows indicate the tendency of the region to either potentiate, remain stable, or depress during the session. Error bands correspond to the mean values ±sem. Source data are provided as a Source Data file.

potentiation in the hippocampus for about half of the recordings in four different animals. Presumably, the variability in the correlation coefficients observed here can be explained by the variability in implantation sites (which is consistent with the fact that no such effect was observed at cortical sites). In any case, the LFP–CBV coupling reported here is of a different nature and occurs on a longer timescale (min) than what was observed in previous studies[47,51,52] and demonstrates that subtle changes in high-frequency electrophysiological activity co-vary with the strong vascular reshaping observed in distributed brain networks.

## Discussion

In this study, we have shown that neuronal and vascular activations associated with natural locomotion involve a brain-wide network composed of the dorsal thalamus, retrosplenial cortex and dorsal hippocampus, but excluding the primary motor cortex which was strongly suppressed. Brain hemodynamics in those regions peaked earlier in the dorsal thalamus and retrosplenial cortex, dentate gyrus and finally CA regions, with a consistent 250–300 ms delay between dentate gyrus and CA regions. We also demonstrated that brain hemodynamics were strongly reshaped between early and late runs within the same recording session. Some regions depressed rapidly after 10–20 runs (cortex), while others were gradually potentiated (CA regions), leaving others relatively unaffected (striatum, ventral thalamus, hypothalamus). These robust patterns occurred while behavioral parameters and theta rhythm were constant and could only be accounted for, in our experimental setup, by hippocampal high-frequency oscillations in the hippocampus.

### Signal measured by fUS and link with single-vessel hemodynamics and neural activity. Previous studies have established

that the fUS signal arises from ultrasound echoes generated by echogenic particles moving at different speeds along different orientations over a typical time window of 200 ms (the duration of a rat's cardiac cycle)[53,54]. In practice, the fUS signal is influenced by multiple factors: number and size of vessels contained in a voxel, vessel orientation, scatterers' velocity, and variations in vessel diameter. These parameters are not easily accessible and influence each other, meaning that absolute values of CBV and CBF are difficult to estimate. However, once a baseline image has been acquired, the number, size, and orientation of vessels can be considered constant. Upon local vasodilation or constriction, only red blood cell (RBC) speed and changes in vessel diameter influence the fUS signal. These two parameters (RBC speed and diameter change) affect the Doppler spectrum differently: 1 – Variations in RBC velocity shift the mean value of the Doppler spectrum (leaving its global power unaffected), a parameter used to build Color Doppler images. 2 – Variations in vessel diameter change the number of scatterers inside a voxel and directly increase or decrease the full energy of the Doppler spectrum, a parameter used to build Power Doppler images. Importantly, this is the case because most vessels are smaller than the size of a voxel, otherwise vessel diameter changes would affect several voxels, but not a single one.

A recent study using concurrent recordings of neural activity, individual blood vessel dynamics and functional ultrasound signal in vivo demonstrated that fUS signal can be predicted from calcium recordings and single vessel hemodynamic through robust transfer functions in a wide set of stimulation paradigms, thus establishing the neural and vascular underpinnings of the fUS signal[55]. This nicely complements another study[56] by the same group where they acquired fMRI signal, fUS signal, and single-vessel dynamics in the same animal in response to different odors. In particular, they found short time lags (on the order of

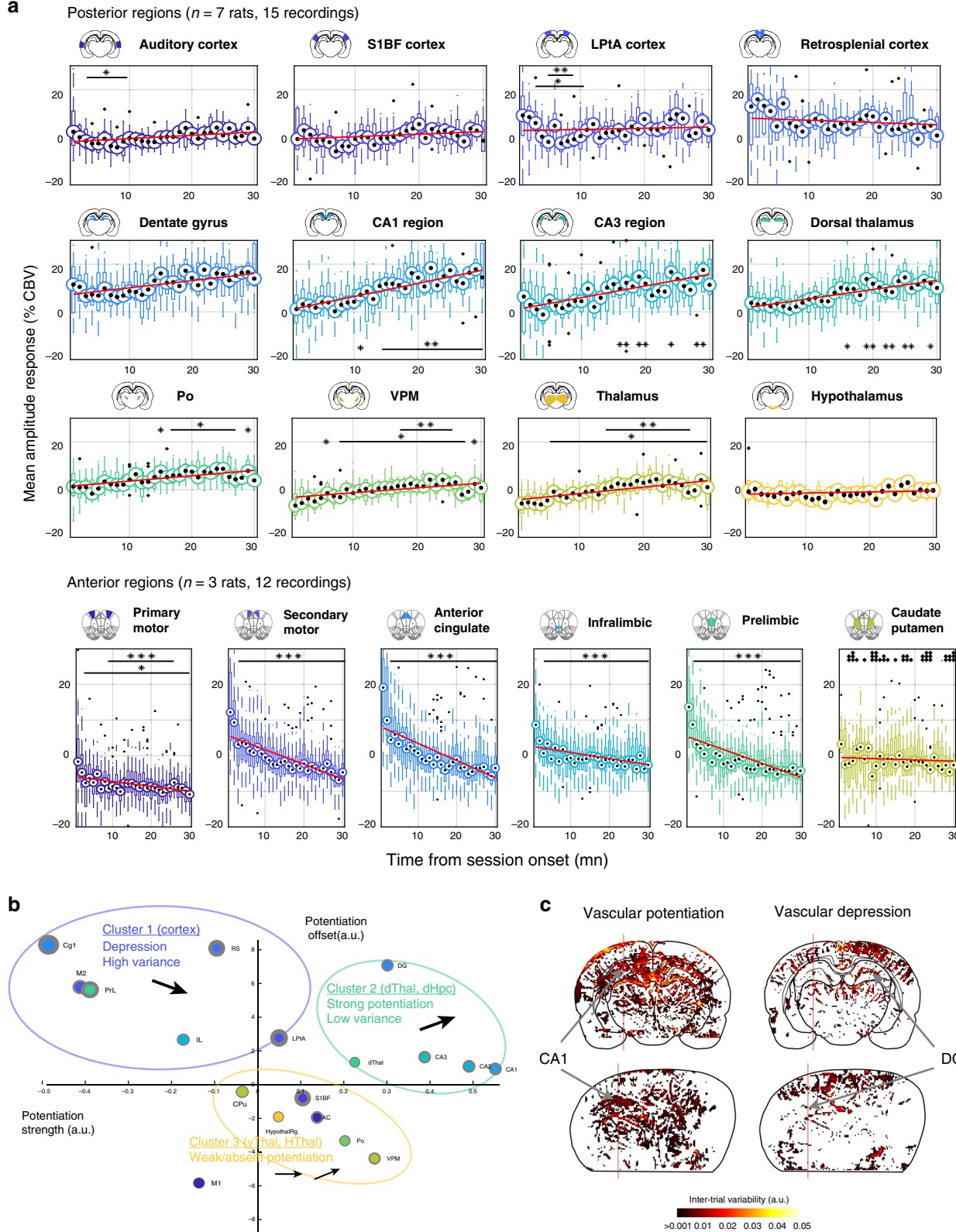

the second) for vascular responses measured by two-photon laser-scanning microscopy (TPLSM) and fUS, whereas BOLD responses were found to be slower and peaked later on the order of tens of seconds. These two studies, though limited to the olfactory bulb of anesthetized animals, clearly establish the transfer functions between neural activity and single-vessel hemodynamics and the fUS signal. Additional work is required to establish the same transfer functions in other structures, ideally in awake animals.

**Linkage with fMRI/electrophysiology studies and astrocytic dynamics.** Simultaneous BOLD fMRI studies and electrophysiological recordings have considerably brought forward our

**Fig. 6 Inverse modulation patterns and temporal dynamics between cortical and hippocampal/thalamic sub-regions. a** Temporal evolution of the regional CBV responses to locomotion from session start (0 min) to session end (30 min) for 12 regions across posterior recording sections (AP = −4.0 mm, $n = 11$ animals, 25 recordings, 384 trials) and for seven regions across anterior recording sections (AP = +3.0 mm, $n = 3$ animals, 12 recordings, 928 trials). All trials were grouped in 1-min bin according to their timing from session onset and we extracted the mean value of the vascular response in a 200-ms window centered on the end of the trial. Whisker plots are displayed for each time bin with temporal linear regression (red line). Note that cortical responses are highly variable and show a marked decrease after the first 3–5 min, both in the anterior and posterior cortical regions. The primary motor cortex (M1) was also depressed but less strongly than the secondary motor cortex (M2) and other frontal cortices. Hippocampal regions and dorsal thalamus in contrast, display a linear dependence on trial timing, with the strongest increase found in the CA1 region. Other regions such as ventral thalamus, hypothalamus, or striatum showed weak to absent modulation across episodes. We tested the statistical significance of the mean distributions between all temporal bins relative to first trial (two-tailed Wilcoxon test, *$P < 0.05$, **$P < 0.01$, ***$P < 0.001$). Whisker plots show the median (center), 25th and 75th percentiles (boxes) and 1st and 99th percentile (whiskers). **b** Clustering of brain regions relative to their vascular potentiation profile. Linear regression-derived potentiation strength (slope in **a**) versus potentiation offset (y-intercept in **a**) for all 18 brain regions. Gray circle lines represent 95% confidence intervals for these two parameters. Three main clusters appear in this representation: cortical regions show a weak linear dependence on trial time, with a marked decrease after early trials (Cluster 1). Dorsal hippocampus and dorsal thalamus display a linear increase in vascular response from early to left trials, with the strongest effect in the CA regions of the hippocampus (Cluster 2). Ventral thalamus, auditory cortex, and hypothalamus show weak to absent vascular potentiation (Cluster 3). Interestingly, M1 did not belong to any of these clusters. **c** Spatial maps of vascular potentiation profile for all pixels in two typical recordings. The color-code is proportional to potentiation strength (slope in **a**) for all CBV pixel signals in a coronal (top) and a diagonal recording (bottom). Positive (left) and negative (right) potentiation slopes are displayed onto two different graphs. Note the difference between cortical regions on the one hand and dorsal hippocampal regions on the other hand, illustrating the two first clusters in **b**. Source data are provided as a Source Data file.

understanding of in vivo neurovascular coupling in the neocortex[57,58]. Fewer studies have examined neurovascular coupling in the hippocampus in rodents and non-human primates due to its limited optical access, but the use of optical fibers and more recently genetically encoded indicators have allowed for the precise interrogations of optogenetically-triggered single-vessel contribution to fMRI signals and for the detection of spreading depression-like events[59]. Dorsal and ventral stimulation of CA1 evoked different brain wide responses[60] and long-term potentiation (LTP) events were clearly detectable in high-resolution fMRI[61].

Importantly, lag times between fUS and electrophysiological signals[47] and between fUS and behavioral cues[62] have been precisely measured and relate tightly to timings measured using optical techniques[28,43,52,63]. We reconciliate the apparent discrepancies in terms of lag times (shorter using intrinsic optical imaging, TPLSM, fUS than the ones found in BOLD studies) arguing that these techniques do not measure the same parameters and therefore do not have access to the same temporal dynamics: BOLD imaging measure a ratio between deoxyhemoglobin and oxyhemoglobin[64], a parameter that has much slower dynamics, than cerebral blood volume (CBV), which dynamics are faster and can be a confounding factor in BOLD studies. Again, this has been recently established in terms of calcium recordings, TPLSM, fUS, and fMRI in the same animal[56].

Last, locomotion has long been known to trigger global increases in astrocytic processes both in the cerebellum and visual cortex[24,25]. These large astrocytic activations were modulated by norepinephrine and were not triggered by light stimulation. Because of the strong involvement of astrocytes in monitoring cerebral perfusion[65] and in neurovascular coupling in general[36,37], a coupling between astrocytic activation and hemodynamic responses during locomotion is probable. We thus expect these two responses to strongly co-vary, but astrocytic activation has been associated with both negative and positive BOLD fMRI signals, leaving the question open[66]. Performing hemodynamic measurements together with astrocytic imaging in head-fixed animals during a virtual reality locomotion test could settle this key question.

**Complex pattern of brain activity during running & motor cortex inhibition.** Several studies have investigated brain hemodynamics during locomotion and reported equivocal results. In head-fixed mice running on a spherical treadmill, somato-sensory cortex showed a strong coupling with gamma oscillations (40–100 Hz) and multi-unit activity, while the adjacent frontal cortex (here primary motor) displayed prominent electrical activity with absent or suppressed CBV response[28]. We reproduced this important result in rats during natural locomotion and observed a marked decrease in primary motor cortex relative to baseline. Arterial and venous blood have been shown to contribute differently to the hemodynamic response function[67] and locomotion was found to drive cortex-wide increases in blood oxygenation – a parameter that fUS imaging cannot capture – and that this effect is mediated by respiration[30]. Our findings are partially in line with autoradiography studies in rats performing treadmill running which found changes in cerebral blood flow in distributed regions, namely dorso-lateral striatum (effect visible on some recordings in our dataset), M1 and M2 motor cortices and cerebellum (not imaged here), but lacked the timing of such activations[31]. These studies rely on the use of treadmills or similar setups which generate different patterns of electrical activity than the ones observed in natural movement, possibly because of the discrepancies between vestibular and visual sensory inputs[68]. In our experimental condition, brain activations can be associated with sensory/visual stimulation (dorsal thalamus), spatial memory processing (dorsal hippocampus, retrosplenial cortex), motor control (primary motor cortex), or reward uptake (Po, VPM, striatum, M1). These are intertwined components, hence, a different experimental design – such as a comparison between reference motor-only trials and goal-directed trials, for example in a T-maze task – is required to disentangle each of these components. Diagonal plane recordings revealed dichotomies within anatomical structures: we found different CBV profiles between ventral and dorsal hippocampus which are known to process spatial information differently[69], between ventral and dorsal thalamic regions, and between anterior and posterior cortices. It is important to keep in mind that animals were overtrained for this task and that they were extremely familiar with the environment, meaning that dorsal hippocampus activations might not be associated with a pure memory component. Last but not least, primary motor cortex showed a peculiar pattern of inhibition during locomotion followed by a marked activation when the animal was taking its reward. This can be explained by the fact that motor cortex seems more implicated in

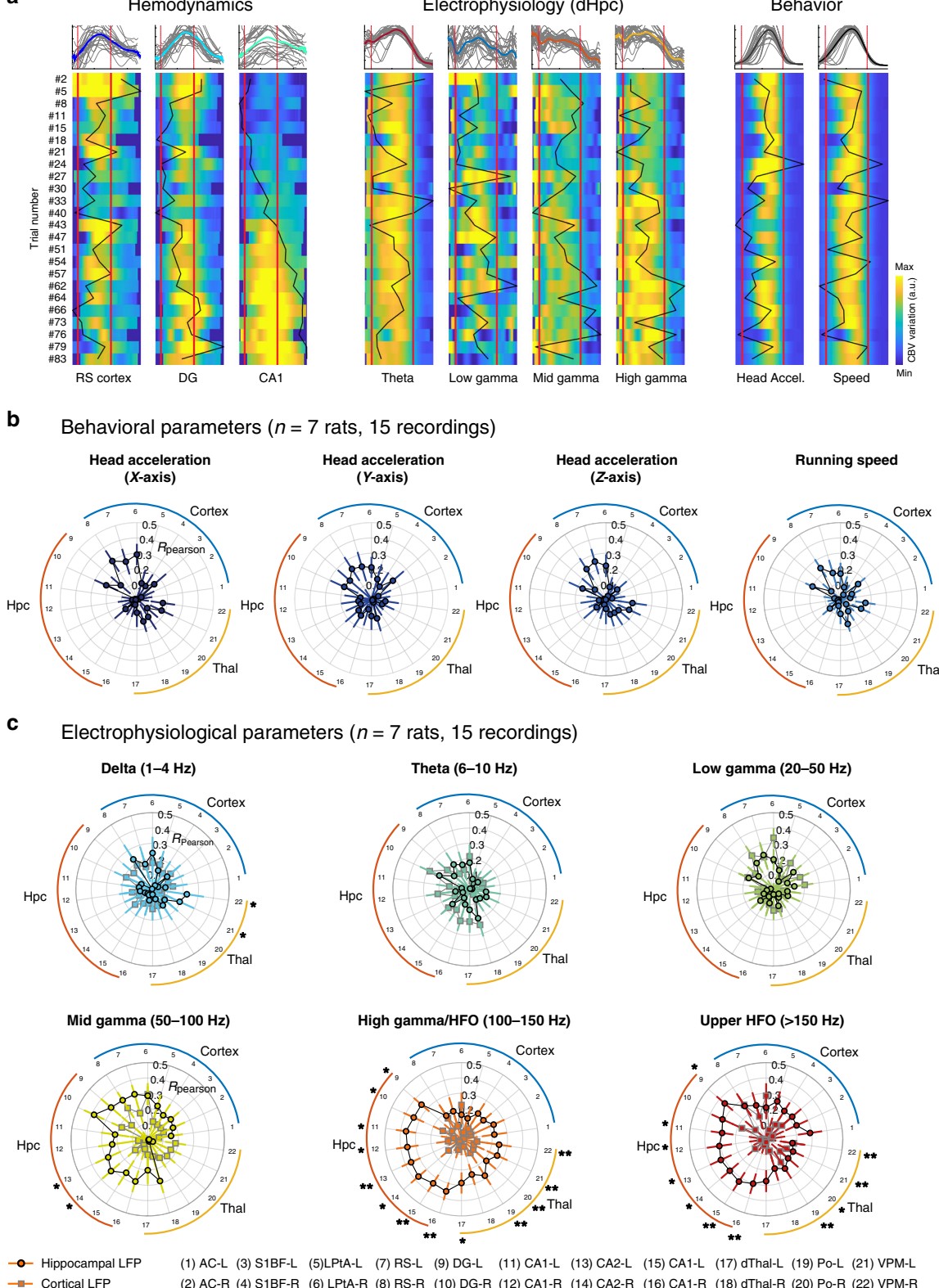

withholding movement and suppressing irrelevant behavior[49] than actually eliciting locomotion which is controlled by a central pattern generator in spinal circuits[70]. Indeed, a large body of evidence showed suppressed neural activity during locomotion in rodents in particular during routine behavior which was the case here[48]. It is interesting to note that M1 was also modulated across trials but less than higher order regions like M2, infralimbic and

prelimbic structures, suggesting that the position of a region in the executive hierarchy might influence hemodynamic reshaping.

**Dynamic sequence of activation in hippocampal subfields.** Our recordings reveal a precise sequential pattern of vascular activation that originated in the dorsal thalamus and retrosplenial

**Fig. 7 Hemodynamic modulation is overall independent of behavior and theta rhythm, but correlates with hippocampal fast gamma/HFO. a** Parallel display of simultaneous CBV signals in three regions (CA1 region, dentate gyrus, retrosplenial cortex) LFP power in four frequency bands (theta: 6–10 Hz, low gamma: 20–50 Hz, mid gamma: 50–100 Hz, high gamma: 100–150 Hz) and two behavioral parameters (speed, transverse head acceleration) for one recording session. Each line is a trial with its corresponding number on the left. CBV variation, LFP power, speed and acceleration power are all scaled to their minimal and maximal value over the full recording and color-coded for comparison. Trials have been re-aligned and temporally rescaled (stretched or compressed) to the same duration (4.2 s), white lines correspond to trial start and end. Black curves give the median value for each trial. Inter-trial correlation is given by the correlation between the black curves for all LFP–CBV pairs and all behavior–CBV pairs. Note the strong linear potentiation in CA1 that correlated only with high gamma power while speed, acceleration, and theta rhythm remain roughly constant. **b, c** Inter-trial correlation analysis of CBV potentiation patterns ($n = 7$ rats, 15 recordings). For each recording, we analyzed whether behavioral parameters (top) and electrophysiological parameters (bottom) co-varied with CBV signals, to account for vascular potentiation patterns observed during locomotion. We thus computed a correlation coefficient for each pair between 16 behavioral/electrophysiological variables (Head acceleration (3), Running speed (1), Band-filtered cortical (6), and hippocampal (6) LFP) and 24 fUS regional hemodynamic signals. This leads to a distribution of correlation coefficients for each LFP–CBV pair and we display the mean (across recordings) on polar plots: a strong correlation coefficient means that a given parameter co-varies (hence can explain) the variability across trials (hemodynamic modulation). Note that neither running speed or head acceleration account for the hemodynamic modulation in any brain region (except for low correlations in the retrosplenial cortex), nor does theta (6–10 Hz), low-gamma (20–50 Hz), or mid-gamma (50–100 Hz). However, both high gamma/HFO (100–150 Hz) and upper HFO (>150 Hz) correlate with vascular signals, especially in the hippocampal and thalamic sub-regions, showing that fast oscillations mirror (hence may explain) the vascular modulation observed across trials. Importantly, cortical LFP did not yield significant correlation patterns meaning that this effect is specific to hippocampal rhythms. For **c** we display the results of statistical testing between cortical and hippocampal Pearson distributions (two-tailed Wilcoxon test, * $P < 0.05$, ** $P < 0.01$). Error bars correspond to the mean values ±sem of the $R_{max}$ distribution (computed using Fischer transformation). The list of exact $p$-values for the three last electrophysiological parameters (mid gamma, high gamma, upper HFO) is given for all 22 sub-regions below: AC-L: 0.2802/0.1029/0.0508, AC-R: 0.5053/0.2603/0.161, LPtA-L: 0.3462/0.2802/0.3462, LPtA-R: 0.4763/0.2603/0.1237, S1BF-L: 0.2234/0.3953/0.0508, S1BF-R: 1.0000/0.5053/ 0.5972, RS-L: 0.1354/0.3012/0.5657, RS-R: 0.5053/0.8722/ 0.8362, DG-L: 0.1129/0.01400/0.0366, DG-R: 0.0695/0.0409/0.1753, CA1-L: 0.1237/0.0140/0.0409, CA1-R: 0.1611/0.0291/0.0258, CA2-L: 0.0291/ 0.0035/0.0108, CA2-R: 0.0409/0.0258/0.0180, CA3-L: 0.0508/0.0035/0.0063, CA3-R: 0.0628/0.0041/0.0082, dThal-L: 0.3953/0.0123/0.0628, dThal-R: 0.1478/0.0935/0.3703, Po-L: 0.1354/0.0022/0.0035, Po-R: 0.1478/0.0072/0.0108, VPM-L 0.3012/0.0063/0.0063, VPM-R: 0.1753/ 0.0063/0.0047. Source data are provided as a Source Data file.

cortex and spread along the sub-regions of the dorsal hippo-campus. The high-definition analysis of these delays revealed 250–300 ms lag between dentate gyrus and CA1 peak response. These are too large to reflect the feed-forward excitation of the tri-synaptic circuit observable, for example, during in vitro 5 Hz stimulation of the performant pathway in deafferented mice hippocampal slices. These so-called 'tri-synaptic circuit waves' last between 60 and 80 ms, corresponding to 5-fold faster time-scale than what we observed[71]. It seems also unlikely that these patterns reflect electrical waves that propagate along the septo-temporal pole of the hippocampus as their traveling speed of 0.1–0.15 m/s is also quite high[6]. However, the fact that hippo-campal subfields hemodynamic responses differ not only in their onset time but also in their temporal profile, with dentate gyrus activity starting earlier and being more sustained when CA1 activation is steeper and more transient, suggests that brain hemodynamics represent population activity first restricted to dentate gyrus before spreading to CA1/CA3 regions, about 300 ms later. This could correspond to an excitatory stimulation of the performant path in CA3 that triggers local inhibition in the CA3 network to contain excitation via interneurons activation[72]. This interpretation also fits well with the idea that CBV signals reflect faithfully the cost of local inhibition[35]. Consistent with this idea, LTP induction via performant pathway stimulation pro-duced detectable changes in BOLD-fMRI signals within hippo-campal subfields in rats, meaning that vascular reshaping could be a proxy of local synaptic plasticity processes[61].

**Region dependence and frequency specificity of neurovascular interactions.** In this study, we have analyzed the LFP–CBV coupling both locally and across distant brain regions. Importantly we found that theta activity is coupled to CBV both locally but also across brain structures, which is consistent with observations that theta travels through the brain and is recorded in many brain structures. Second, we reproduced an important finding in the literature showing that neurovascular coupling is highly region

dependent. In particular, we found a reverse relationship at the primary motor cortex sites where robust gamma oscillations (50–100 Hz) are found during locomotion, associated with a reduction in the CBV signal. This is in line with the literature showing that motor cortex is often suppressed during repetitive behavior[49] and that there is a decoupling between neural and vascular signal in the motor cortex of head-fixed running mice[28]. Thus, the absence (or presence) of robust LFP signal cannot be equivocally associated with a reduction (or increase) in CBV signal but shall be studied in every region independently. In the future, the mechanisms of the adaptive vascular responses may have to be explored differentially across regions.

Previous studies that recorded concurrently cerebral hemody-namics and local field potentials have shown that fast gamma oscillation strongly correlates with subsequent vascular signals[52,58,73]. We recently showed that during REM sleep, phasic bouts of high gamma oscillations strongly correlate with subsequent vascular activity in almost all brain regions[47] (theta oscillations also, but more moderately). During locomotion, however, theta rhythm – that is linearly related to speed – correlated better with vascular activity than mid and high gamma in all brain regions. Additionally, neurovascular coupling shows a stronger regional dependence: whereas REM-sleep hemody-namics activations were hypersynchronous across brain regions, revealing very strong LFP–CBV coupling in most regions, locomotion-related hemodynamics were restricted to a subset of precise (but distributed) regions. This means that neurovascular coupling is not only region-dependent but also state-dependent. When assessing CBV changes on slower timescales, we found that high-gamma/HFO correlated with CBV in the dorsal hippocam-pus and thalamus on a slower timescale. A lead explanation on why low gamma, mid gamma and high gamma/HFO relate differently to CBV signal is that they involve different cell types and populations, hence require different energy budgets, that may or may not trigger detectable changes in vascular activity.

If theta-gamma coupling in the hippocampus could be measured accurately in this experimental condition, we could

investigate whether the coupling strength (as measured by a modulation index) co-varies with brain hemodynamics and whether it correlates better with CBV than the power of fast gamma oscillations per se. Indeed, increased power of gamma oscillations does not necessarily entail stronger phase-frequency coupling and conversely. If the coupling strength did correlate with CBV, one leading explanation might be the increased interaction between fast-spiking interneurons and principal cells with changes in synaptic plasticity. It remains to be determined how theta and faster gamma are coupled mechanistically and whether they are mediated by small changes in the activity of another independent group of neurons, or by local excitatory and inhibitory interactions that may not necessitate significant changes in energy demand, hence would not yield to detectable blood flow changes.

**'Online' modulation of brain activity across trials despite stereotyped behavior.** Similar patterns of hemodynamic modulation, especially in the mean CBV level over slow (minute) timescales, have been reported in monkeys performing a visual task and were related to reward and to the degree of engagement in the task, high-reward trials eliciting a decrease in global CBV and low rewards showing an opposite effect[74]. This means that hemodynamic reshaping observed here could be related to longer pauses between rewards in the late part of the running task, but as animals took longer breaks towards the end of the session, CBV levels increased in the dorsal hippocampus which is the inverse effect of that seen in the cortex of monkeys. Additionally, hemodynamic modulation in the cortex occurred over a rapid timescale (in the first 5–10 min of the session) during which time between reward and running speed were remarkably constant. Importantly, hemodynamic modulation was stronger in the dorsal hippocampus than in any other region, especially in the CA1 region. Rats are quick to learn new environments and stabilize an internal representation of space only after minutes of exploration in new mazes[75]. In our view, hemodynamic reshaping reflects ongoing adaptive processes at play during stereotyped running. It is unlikely to be related to the asymmetric expansion and tuning of hippocampal place fields occurring during repeated locomotion behavior[76], because place fields are formed rapidly even in a new environment and backward shifts are reduced after a dozen trials, which in our setup occurs in the first 3 min of the session. Another explanation could be that vascular networks are reorganized online during the task to reinforce relevant information location. Such reorganization of vascular networks has been shown to occur on very fast timescales to support the transfer of long-term memories from hippocampal to cortical sites in an odor-recognition task[77]. In a similar view, hemodynamic reshaping could mirror the locomotion-dependent remapping of distributed networks over prolonged periods of time[27]. Last, we observed that hemodynamic potentiation in CA1 precedes high-gamma potentiation by a few trials: one explanation could be that vascular activity directly or indirectly modulates neuronal activity in the subsequent trials, for example, through astrocytic modulation or temperature modifications[78].

**Mechanisms that could drive hemodynamic reshaping.** Several important parameters could modulate brain hemodynamics during locomotion including heart rate, respiration, or temperature. Controlling all of these parameters independently remains to be done in the future, but we can already rule out or at least lower the probable influence of some of these parameters based on the spatial profile of hemodynamic reshaping. A major homeostatic effect arising from increased heart rate, respiration, or neuro-modulation is likely to affect all brain structures similarly: we

would probably not observe a decrease in the cortex, concurrent with an increase in the CA1 region and a stable response in the dorsal thalamus if heart rate was the main driver of hemodynamic reshaping. Also, these parameters are directly related to running speed which remained constant across trials.

Brain temperature however is known to affect brain activity significantly[79]. Endogenous heat arising from neuronal activity and muscular effort can modulate both neuronal firing rate and hemodynamic signals[80,81], in particular, over the accumulation of trials. This could also happen exogenously from ultrasound-insonification, but because we use a sequence that imposes a 40-s refractory period (resulting in a 20–25% efficacy in imaging time) and the fact that our animals underwent large craniotomies should result in a substantial dissipation of heat in brain tissue[82]. Also, neighboring structures like dorsal thalamus and ventral thalamus, or dentate gyrus and CA1 region show strong discrepancies in their modulation patterns, thus if we cannot completely exclude an effect of temperature, it probably super-imposes on other underlying causes of vascular reshaping. Another possible explanation of the remarkable lasting increase in CBV in the hippocampus is that local control of vasculature may be region-specific and that feedback regulation in the hippocampus may be wired in such a way to promote such a slow 'recovery' back to baseline which would favor supporting long-lasting 'plastic' event. Though, it is hard to imagine any plastic event occurring in such a stereotypical task (running in a linear maze), the hippocampus, surprisingly, may be wired in this manner.

We have performed fUS imaging during the recovery period from the isoflurane anesthesia, which showed a marked decrease in all brain regions when anesthesia was stopped (Supplementary Fig. 9). Animals usually recovered from anesthesia within minutes, but the fUS signal needed an additional 5–10 min to go back to baseline depending on the brain region. We let the animal recover for another 20 min and checked visually that the fUS patterns were stable and resembled those observed during wake. This is a first argument to rule out a possible implication of the anesthesia in the modulation of hemodynamic patterns observed. Another argument follows from the fact that though the anesthesia duration was fairly comparable across sessions (15–20 min), the actual start of the running session was quite variable across recordings and showed no clear correlation with the potentiation patterns observed in the hippocampus. From our perspective, it is unlikely that such delay would affect vascular potentiation, though this remains to be checked experimentally. The fact that, at least in the hippocampus, CBV changes were mirrored by increased power in high-frequency oscillations, while these remained stable at the cortical sites supports the idea of local vascular plasticity in the hippocampus.

The final question is the possible function(s) of such hemodynamic reshaping if any. A key element is the gradual disengagement of the cortex to the benefit of subcortical structures, which aligns with the animal's behavior: during early trials, the animal senses and sniffs its environment, probably recognizing it and re-exploring it for a bit. Then it turns into 'automatic' mode and engages in repetitive behavior, when in parallel subcortical structures 'discharge' cortical ones and display stronger individual responses. Introducing sudden changes (unexpectedness) in reward distribution and investigating concurrent hemodynamic modulation could help test this hypothesis.

Our data suggest that the stereotyped repetition of locomotion endogenously modulates brain hemodynamics on a rapid (minute) timescale, and does so without noticeable behavioral modification. One of the conclusions of our study is thus that inter-trial averaging must be performed with caution (for example, over a small number of trials). Average hemodynamic

response functions, commonly derived in neuroimaging studies, might totally erase trial variability or worse: perform averaging between very dissimilar activations. Indeed, two seemingly similar repetitions of a given behavior may actually differ totally in terms of brain activity. Secondly, it shows that neurovascular interactions are complex and couple LFP recording and CBV regional signals over different timescales: though hippocampal theta rhythm correlate with hippocampal vascular activity at the second timescale, it does not account for the slower modifications of CA1 hemodynamics across trials, which are massive. Probing the activity of brain hemodynamics at large scale and comparing them to local neuronal activity monitored with electrodes is a work in progress that will considerably help us understand the physiological basis of complex behavior.

## Methods

**Animal surgery**. All animals received humane care in compliance with the European Communities Council Directive of 2010 (2010/63/EU). The experimental protocol used in this study was extensively reviewed and approved by the French CEEA (Comité Ethique pour l'Expérimentation Animale) n°59 Paris Centre et Sud under the reference 2018061320381023. Adult Sprague Dawley rats aged 10–12 weeks underwent surgical craniotomy and implant of an ultrasound-clear prosthesis. Anesthesia was induced with 2% isoflurane and maintained with keta-mine/xylazine (80/10 mg/kg), while body temperature was maintained at 36.5 °C with a heating blanket (Bioseb, France). A sagittal skin incision was performed across the posterior part of the head to expose the skull. We excised the parietal and frontal flaps by drilling and gently moving the bone away from the dura mater. The opening exposed the brain between the olfactory bulb and the cerebellum, from Bregma +6.0 to Bregma −8.0 mm, with a maximal width of 14 mm. Electrodes were implanted stereotaxically and anchored on the edge of the flap. A prosthetic skull was sealed in place with acrylic resin (GC Unifast TRAD), and the residual space was filled with saline. We chose a prosthesis approach that offers a larger field of view and prolonged imaging condition over 4–6 weeks compared to the thinned bone approach. The prosthetic skull is composed of polymethylpentene (Good-fellow, Huntington UK, goodfellow.com), a standard biopolymer used for implants. This material has tissue-like acoustic impedance that allows undistorted propagation of ultrasound waves at the acoustic gel-prosthesis and prosthesis–saline interfaces. The prosthesis was cut out of a film of 250 μm thickness and permanently sealed to the skull. Particular care was taken not to tear the dura to prevent cerebral damage. The surgical procedure, including electrode implantation, typically took 4–6 h. Animals were injected with anti-inflamatory drug (Metacam, 0.2 mg/kg) prophylactic antibiotics (Borgal, 16 mg/kg) and postoperative care was performed for 7 days. Animals recovered quickly and were used for data acquisition after a conservative one-week resting period.

**Electrode design and implantation**. Electrodes are based on linear polytrodes grouped in bundles of insulated tungsten wires. The difference with a standard design is a 90°-angle elbow that is formed prior to insertion in the brain[44]. This shape enabled anchoring of the electrodes on the skull anterior or posterior to the flap. Electrode bundles were implanted with stereotaxic positioning micromotion and anchored one after another. The prosthesis was then applied to seal the skull. Two epidural screws placed above the cerebellum were used as a reference and ground. Intra-hippocampal handmade electrode bundles were composed of 25–50 μm diameter insulated tungsten wire soldered to miniature connectors. Four to six conductive ends were spaced 1 mm apart and glued to form 3-mm-long, 100–150-μm-diameter bundles. The bundles were lowered in the dorsal hippo-campi at stereotaxic coordinates AP = −4.0 mm, ML = ± 2.5 mm, and DV = −1.5 mm to −4.5 mm relative to the Bregma. Before each surgery, the relative position and distance between each recording site on the bundle (4 to 6 recording sites per bundle) was identified by measuring the impedance change, while low-ering the electrodes in saline solution (Na-Cl 0.9%). Electrodes sites' locations were cross-validated post mortem via electrolytic lesions and histology was performed to reconstruct the tract of electrode bundles in the tissue. The actual design was based on handmade electrodes with minimal spacing of 500 microns between recording sites and a maximal number of 6 electrodes per bundle. This allows to observe the characteristic phase reversal between the superficial and deep layers of the dorsal hippocampus[83], but not to quantify the cross-frequency coupling as with linear probes. Additionally, the surgical procedure is complex and there is variability between targeted structure and actual electrode position due to brain tissue movement (swelling) both during and after the surgery. See Supplementary Fig. 1 for further details.

**Recording sessions**. Eleven male Sprague Dawley were trained daily to run back and forth on a 2.4-m linear track for a water reward given on both ends through solenoid valves. The rats were placed under a controlled water restriction protocol (weight maintained between 85 and 90% of normal weight) and trained for 5 days

before surgery. The track (240 × 20 cm) had 5-cm-high lateral walls and was placed 50 cm above the ground. Each time the animal crossed the middle of the track, one drop of water was delivered in alternation through water tubes by opening an electronically controlled pair of solenoid valves. Daily training lasted 30 min. Rats then underwent a surgery combining implantation of handmade linear LFP probes in the dorsal hippocampus with permanent skull replacement by a polymer prosthesis allowing undistorted ultrasound passage. At the start of each recording session, to attach the ultrasound probe and connect the EEG, the rats underwent brief anesthesia for 15–20 min with 2% isoflurane. Acoustic gel was applied on the skull prosthesis, and the probe was inserted into the probe holder. The gel did not dry out even for extended recordings of up to 6–8 h. The animals were allowed to recover from anesthesia for 40–60 min before starting the recording session (see Supplementary Fig. 9). A typical session included a 30–40 min running period. In total, we recorded from 11 animals over the coronal and diagonal planes for a total of 42 running sessions and from three animals over the frontal plane for a total of 15 recording sessions. Though overall distance traveled was lower in fUS-EEG animals than in controls, our protocol did not impede instantaneous animal movement (no significant difference in peak speed). This difference can be explained by the higher load onto the animal's head, resulting in longer inter-trial interval, probably due to longer rest periods[44].

**LFP acquisition**. LFP, electromyogram (EMG), and accelerometer (ACC) signals and video were monitored continuously from video-EEG device for offline pro-cessing. Intracranial electrode signals were fed through a high input impedance, DC-cut at 1 Hz, gain of 1000, 16-channel amplifier and digitized at 20 kHz (Xcell, Dipsi, Cancale, France) for half of the recordings and through a Blackrock Cereplex System using the Cereplex Direct Software Suite (version 7.0.6.0) developed by Blackrock Microsystems (Salt Lake City, UT, USA) for the second half, together with a synchronization signal from the ultrasound scanner. LFP signals were pre-amplified and digitized onto the animal's head which prevent artifacts originating from cable movement. Custom-made software based on LabVIEW 2016 (National Instruments, Austin, TX, USA) simultaneously acquired video from a camera pointed at the recording stage. A regular amplifier was used, and no additional electronic circuit for artifact suppression was necessary. A large bandwidth amplifier was used, which can record local field potentials in all physiological bands (LFP, 0.1–2 kHz). The spatial resolution of LFPs ranges from 250 μm to a few mm radius.

**Ultrasound acquisition**. Vascular images were obtained via the ultrafast com-pound Doppler imaging technique[84]. The probe was driven by a fully program-mable GPU-based ultrafast ultrasound scanner Aixplorer for the first half (Supersonic Imagine, Aix-en-Provence, France) and Verasonics (Kirkland, USA) for the second half, both relying on 24 Gb RAM memory. We acquired 6000 ultrasound images at 500 Hz frame rate for 12 s, repeating every 40–60 s (refractory period of 40 s during which no data can be acquired). Each frame is a compound plane-wave frame, that is, a coherent summation of beamformed complex in phase/quadrature (IQ) images obtained from the insonification of the medium with a set of successive plane waves with specific tilting angles[85]. This compound plane wave imaging technique enabled the re-creation of a dynamic transmit focusing at all depths a posteriori in the entire field of view with few ultrasound emissions. Given the tradeoff between frame rate, resolution, and imaging speed, a plane-wave compounding using five 2°-apart angles of insonification (from -5° to +5°) was chosen. As a result, the pulse repetition frequency of the plane wave transmissions was 500 Hz. To discriminate blood signals from tissue clutter, the ultrafast com-pound Doppler frame stack was filtered, removing the 60 first components of the singular value decomposition, which optimally exploited the spatiotemporal dynamics of the full Doppler film for clutter rejection, largely outperforming conventional clutter rejection filters used in Doppler ultrasound[86].

**LFP analysis**. All analysis was performed in MATLAB (version R2017b, Math-works, USA). NPMK package (version 4.5.3.0) developed by Blackrock Micro-systems was used to import the raw LFP data into MATLAB. EEG was collected and pass-band filtered between 0.5 and 1 kHz. In order to remove movement artifacts, we computed differential LFP signals by subtracting the raw signal from two neighboring sites, based on the position of each electrode along the bundle. This, together with direct amplification onto the animal's head via the INTAN chip from the Blackrock system, allows for quality and artifact-free LFP recording in the motor cortex and hippocampus during free running. Though we cannot completely exclude potential contamination of high-frequency LFP recordings by multi-unit and mus-cular activity, we observed that fast gamma oscillations displayed typical phase-amplitude coupling patterns for all recordings robustly across animals. The size of our electrode diameters (25–50 microns) and the stability of our recordings throughout sessions decrease the probability that fast gamma events arose from correlated spiking activity. EEG was first filtered in the LFP range (LFP, 0.1–2 kHz) and band-pass filtered in typical frequency bands including delta (1–4 Hz), theta (6–10 Hz), low-gamma (20–50 Hz), mid-gamma (50–100 Hz), high-gamma (100–150 Hz), and ripple (150–250 Hz). This division has been thoroughly described and proven to be functionally relevant for hippocampal electrographic recordings[50]. The power envelope of LFP oscillations was computed as the square of the raw signal integrated over a sliding Gaussian kernel of a characteristic width of 500 ms.

**Behavior analysis**. Videos were processed frame by frame, by drawing a fixed rectangle to isolate all pixels within the track. We then performed thresholding on the histogram of the pixels inside the track to isolate bright pixels of the rat's body and compute their barycenter. Animal's position is then computed frame by frame and smoothed with a Gaussian kernel (half-width 500 ms). Speed vectors are computed in two dimensions and re-projected into the actual track coordinate to extract relevant body speed. Candidate runs were detected by thresholding speed (threshold: 0.1 m/s) and extracting peak speed, start (ascending threshold crossing) and end (ascending threshold crossing). A run was then validated if the trajectory between the start and end points was monotonous over the long axis of the track, over a minimal distance of 1 m for a maximal duration of 10 s. We then used the start time for trial re-alignment. For Figs. 6 and 7, the CBV (and LFP for Fig. 7) activation for a single was computed as the mean (median for Fig. 7) value in a 200-ms sliding windows centered on the end of each trial, as defined above. This allows a relatively robust evaluation of the activation in a single trial.

**CBV maps & spatial averaging**. Previous studies from our group have shown that the fUS signal tightly relates to neuronal activity and microscopic single-vessel hemodynamics. In order to build the CBV maps from the raw back-scattered echoes, radiofrequency (RF) signals are delayed and summed to form IQ matrices through a process known as beamforming. Theses matrices are then decomposed via Singular Value Decomposition (SVD) to decouple slow movements due to pulsatility and tissue motion from fast movements due to echogenic particles crossing a voxel during a full cardiac cycle (200 ms). Importantly, Power Doppler images are computed by taking the power of the full Doppler spectrum, including a range of speeds in large and smaller vessels, with an inferior bound of 5–10 mm/s in axial velocity. This gives a signal proportional to the number of echogenic particles that have crossed a single voxel during 200 ms (with a sufficient axial velocity) which is a good estimate of local cerebral blood flow (CBF). We thus can build Doppler movies with a sampling frequency of 5 Hz, which can be pushed to the pulse repetition frequency (here 500 Hz) through the use of a temporal sliding window. To derive CBV maps from the raw Doppler movies, we performed voxel-wise normalization from a baseline period: before the animal started the session, we recorded three 12-s 'bursts' during quiet wakefulness while the animal's cage was placed nearby one end of the track. We extracted the distribution for each voxel during this baseline period and computed a mean value, leading to one value for each voxel of the image. To derive a signal similar to ΔF/F in fluorescence microscopy, we subtracted the mean and divided by the mean for each voxel in the Doppler movie. This allowed normalization and rescaling of ultrasound data, yielding to an expression in terms of percent of variation relative to baseline (CBV % change). Each voxel was normalized independently before performing spatial averaging. An important point is that this process is sensitive to movement, so we had to ensure a strong fixation using pressure screws of the probe onto the animal's head during locomotion.

**Atlas registration**. Coronal recordings were registered to two-dimensional sections from the Paxinos atlas[87] using anatomical landmarks, such as cortex edges, hippocampus outer shape, and large vessels below brain surface as a reference. We performed manual scaling and rotation along each of the three dimensions to recover the most probable registration. Once performed, regions of interest were extracted using binary masks. To register diagonal planes (that had no direct coronal correspondence), we segmented each two-dimensional recording plane into anatomical regions based on a 3D MRI-based whole-brain atlas, which provided labeling for 52 brain regions[88]. To derive the anatomical regions of interest, we designed a customized registration algorithm[47], which projected our two-dimensional ultrasound plane onto the three-dimensional volumetric MRI dataset. In short, we manually pinpointed landmarks on the ultrasound image including the outer cortex edges, inner cortex edges, midline plane, and dentate gyrus edges, which were prominent on fUS images. We defined nine parameters, including three offset values, three scaling values, and three rotations (13 parameters for multiplane registration), to identify a given plane unambiguously in the 3D Waxholm MRI space. We then performed optimization using the Simplex method (custom code in LabView 2016) to minimize the global error based on the position of our landmarks and the closest corresponding border in the Waxholm space. Provided the algorithm did not start too far from the actual position, it converged quickly and yielded robust registration for any ultrasound plane, including the diagonal ones. This process allowed us to derive vascular activity in 20 regions of interest that were present in both coronal and diagonal sections (coronal view coordinates Bregma = −4.0 mm; diagonal view coordinates 45° relative to the sagittal midline).

**LFP–CBV correlation analysis**. To assess the association between LFP events and CBV variables, we searched for correlations between each possible combination of LFP band-pass filtered signals and regional CBV variables. As neurovascular processes are not instantaneous, we considered possible delays between electrographic and vascular signals and thus computed cross-correlations functions between the two signals for any LFP–CBV pair and any lag in a given time window (−1.0 to 5.0 s). We performed this analysis over pixel and regional variables, but only regional variables allowed for statistical comparison across recordings.

**Statistics and reproducibility**. All statistics are given as ±standard error of the mean, unless stated otherwise. Statistics in Fig. 3 are computed on $n = 11$ animals over 42 recording sessions. Bar diagrams shown in Fig. 3 are computed by averaging the mean values of 22 recordings on 11 animals for the coronal planes and 20 recordings in seven animals for the diagonal planes. Statistics are computed using a two-tailed Mann–Whitney test. The significance of Pearson correlation coefficients shown in Figs. 3 and 7 are assessed by computing the $t$-value (using $t = \frac{r\sqrt{1-r^2}}{\sqrt{n-2}}$) and reporting it in Student's table with $n$-2 degrees of freedom. Statistical testing for correlation distributions were computed after Fischer transformation. Multiple comparison for regional analyses were accounted for using Bonferroni correction. Due to the difficult experimental constraints (difficult surgical procedure, precise electrode implantation, habituation, and training required for the locomotion task) no replication attempt was performed in this study, but the results were robust and observable across individuals and recordings.

**Reporting summary**. Further information on research design is available in the Nature Research Reporting Summary linked to this article.

## Data availability

All data and software supporting the findings of this study are available from the corresponding authors upon reasonable request. Custom codes used for the collection of fUS data are protected by INSERM and can only be shared upon request, with the written agreement of INSERM. Source data are provided with this paper.

## Code availability

The code used to generate the results that are reported in this study is available from the corresponding authors upon reasonable request. Custom codes used for the analysis of fUS/LFP/video data used in this study are protected by INSERM and can only be shared upon request, with the written agreement of INSERM.

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

## Acknowledgements

The authors would like to thank M. Matei, F. Segura, M. Nouhoum, H. Serroune, A. Bertolo, S. Hubatz, and T. Watson for their help in the different steps of the research leading to these results. The authors would like to thank G. Girardeau, K. Benchenane, and S. Pezet for interesting discussions and advice on the manuscript. The research leading to these results has received funding from the European Research Council under the European Union's Seventh Framework Programme (*FP7/2007-2013*) / ERC *grant agreement* n° 339244-FUSIMAGINE. This work was supported by the AXA Research Fund under the chair *New hopes in medical imaging with ultrasound*. This work was partially supported by the National Institutes of Health (grant U01NS099724).

## Author contributions

A.B. and I.C. designed the experiment. A.B. designed the electrodes, performed the surgeries, training, and recording sessions. E.T. and T.D. programmed the ultrasound sequences and burst recording mode. C.D. designed the clutter-rejection algorithm. I.C. programmed the acquisition software and atlas-registration algorithm. A.B. analyzed the behavioral and electrographic data. A.B. and M.T. analyzed the ultrasound data and discussed multimodal analysis. All authors wrote the paper.

## Competing interests

The authors declare the following competing interests: T.D. and M.T. are co-founders and shareholders in the ICONEUS company.
