## [Peer Review File · Nature Communications]

Reviewers' comments:

Reviewer #1 (Remarks to the Author):

This is a highly innovative and original study investigating the association between cerebral blood flow in different brain regions with or without field recordings of rhythmic activity in freely behaving mice running a linear track. This is a follow-up to their previous publication on the correlations between gamma activity in hippocampus in relation to cerebral blood flow measurements with the same instrument used here, but focused mostly on sleep (REM and NREM). Here, they performed a unique set of experiments using this powerful tool of doppler imaging developed by this group which offers the capacity to record blood flow with high temporal precision and good spatial accuracy. This doppler-based technique is extremely powerful and has several advantages compared to fMRI including much higher temporal resolution and the ability to combine this method with other techniques (here electrophysiology) in freely behaving rats. Certainly, this technique has a bright future.

Particularly, this study investigates the changes in CBV and theta and the gamma frequencies in association with speed in over trained rats running on a linear maze with two rewards on both ends. The recordings of the doppler responses were done coronally and in diagonal planes. The regions showing marked changes in CBV were the parietal cortex, retrosplenial, all of hippocampus regions and dorsal thalamus. Interestingly, the time lag of the CBV in association with the peak speed showed that the thalamus showed a short lag compared to the hippocampus and cortex, with the dorsal thalamus and dentate gyrus showing the shortest time lags. There appears to be a sequence in BV when cross-correlated to speed. Similarly, to their previous study done in sleep, there appears to be a high correlation between theta and high-frequency gamma, and a weak correlation with low-gamma and CBV during running. Some of the most interesting results show that the dorsal hippocampus and the thalamus have increased CBV whereas neocortex tend to show an inverse relation to CBV in relation to continuous running episodes suggesting that these regions could be perfused by different vascular systems.

The strength of this study is to use a combination of CBV measurements and field recordings in a stereotyped linear running task. Several results are novel and interesting, and this study represents a novel characterization of CBV changes recorded simultaneously in different brain areas. However, the changes observed are difficult to interpret and the significance of the results difficult to put into broader context. This study is definitively new and important but I find it hard to understand the implications, but admittedly this is due in part to the fact that there are almost no studies that have investigated this phenomenon in as much detail and moreover in freely behaving animals. Because the results are novel but the interpretation difficult, the authors may wish to clear-up some of the result sections. Comments:

-While the results shown in Figs 1 and 2 are noteworthy, the sequential activation displayed in Fig. 3 is less clear. While overall patterns within regions (ie all hippocampus and all thalamus) appear convincing, the sequential activation (Fig. 3C and D) and its smaller regional distribution of CBV is less convincing. While some sequential distribution appears significant (Fig 3C, left lane) for hippocampus is clear, that on the right lane is not and does not show the same sequence of CBV? Furthermore, the results shown in C don't seem to fit the results for the group distribution in d. However, the distribution of coronal and sagittal results appears comparable. Is there a real distribution of the tri-synaptic loop in hippocampus given the data in d?

-As for the electrophysiological recordings using tungsten wires, it was not indicated clearly where the wires were located (histology?). Obviously, recordings of theta and gamma, as well as their phase-locking values will depend on which layers was recorded in hippocampus. Therefore, its unclear how reliable are the cross-correlation of theta with faster frequency oscillations. Moreover, even if this coupling was measured accurately (using linear probes for example), the interpretation of the relationship between the coupling and the with CBV would likely be difficult to explain. What would be the interpretation?

-Fig 4 shows the relationship between the CBV and the field recordings. The high correlation between theta and high-gamma, but not low-gamma, are expected since previous studies have shown the correlation between these frequencies and running speed per se. Also the data showing that the field recordings preceded the CBV change was also noted in their previous study.

-The results shown of Fig 5 and 6 showing the differential dynamics of the CBV changes in different areas in relation to trial numbers is very interesting and unexpected. However the interpretation is more difficult. The hemodynamic reshaping probably does not reflect changes in the tuning of place fields or plasticity-related shifts in the place cells as cited by the authors since place cells are formed rapidly when running in the same linear maze and stabilize rapidly (backward shifts are reduced after a dozen trials) after a dozen of running trials. Therefore, the timeframe of the increase in CBV in hippocampus observed here and place cells changes do not occur in the same timescale. Anyhow, this issue could be settled by using an NMDA antagonist which are know to block these backward shift changes.

-How can the CA2 area be accurately and correctly be measured since it's a fairly small structure and the area measured by the system appears at the limit?

Reviewer #2 (Remarks to the Author):

Bergel et al. applied a functional ultrasound imaging method to map the hemodynamic responses from a single brain slice (two orientations) with concurrent electrophysiology in the hippocampus of normal behaving rats during running. The overall technological establishment is impressive. However, the experimental design needs to be clarified and the interpretation of results needs to be better justified with good control studies. Also in the discussion section, these authors covered several key observations without a thorough discussion of the literature, leaving the conclusion less convincing. Several major concerns should be dealt with to improve the overall quality of this manuscript before it can be published.

1. This work is done from the pioneering group on fUS led by Dr. Tante. Despite the great track record of the application of fUS to map brain dynamics, it remains hard for the readers to understand what is exactly the signal detected by fUS. The authors claimed the CBV signal measurement with some references. But, the cited references do not show a clear definition or explanation of how CBV signal was measured with fUS. If the functional map is CBV % change. How was it calculated? It should be clearly explained in the method section. It is crucial to specify the signal detected by fUS, which is also critical for the interpretation of the lag times in comparison to the existing fMRI studies. In particular, the authors need to discuss their work in the context of the existing literature of hemodynamic mapping and neurovascular coupling in animals.

a. The authors need to discuss the fMRI studies of the hippocampus with electrophysiology or calcium recordings in non-human primates and rodents. The specific hemodynamic responses have been well described in terms of lag times or correlation features. The very short duration and lag time of the "CBV" signal reported in this manuscript needs better justification.

b. The authors also attributed some of their unique observations to astrocyte function involved in neurovascular coupling, but a better job should have been done to elaborate on how astrocyte function might be a potential cause. During locomotion, there are decent studies to record the astrocyte function in the cerebellum and the cortex in normal behaving mice. Please provide a thorough discussion. Importantly, A series of studies show the global astrocyte activation upon locomotion or arousal state changes. Then, how will astrocyte function be coupled to hemodynamic responses in the brain during locomotion? The authors need to provide better linkage between their observation and the existing literature.

2. The LFP data acquisition. During locomotion, the LFP signal will be possibly contaminated with motion artifacts, the authors need to provide some detailed information on how their LFP signal is avoided from the artifacts or how they process the LFP data. In particular, if no robust fUS signals were reliably observed in some part of cortices. What will be the LFP signal look like? It should be used as negative control data to justify the LFP signal detected in the hippocampus. It is a critical

experiment to justify the existing results.

3. Some positive controls are missing to strengthen the arguments of the work in hippocampus. The authors present some highly variable fUS signals in different brain regions with only LFP recording in the hippocampus (also across trials). Thus, it is crucial to first identify the activation in the motor cortex or cerebellum during locomotion as the starting point to validate their method. Also, the LFP recording of the cortex is a crucial control signal to be presented when interpreting the fUS signal in the cortex during locomotion (refer to point 3).

4. Statistics need to be better clarified. For example in Fig 6 and 7, how many rats were used to make the overall presentation? Are those observations reproducible?

5. The statement of vascular potentiation or plasticity is missing justification. First, based on the method description. The rats were anesthetized for 20-25min and wait for 40min before testing with 30 min duration. How can they be sure that the time-dependent responses are not due to the washout of the anesthetic effect after the rats started "running"? Secondly, do they perform another consecutive round of training from the same animals with varied resting time and test if the increased hippocampal fUS signal remains? Thirdly, a positive control site, e.g. motor cortex or cerebellum, is needed to examine if there is a real "vascular plasticity" by comparing it with the concurrent LFP (increased vascular responses do not mean that there is vascular plasticity.)

Overall, this work shows some promising and interesting aspects of fUS brain mapping of normal behaving animals. The technique is interesting and inspiring, but the scientific arguments of this manuscript need to be improved thoroughly.

Reviewer #1 (Remarks to the Author):

This is a highly innovative and original study investigating the association between cerebral blood flow in different brain regions with or without field recordings of rhythmic activity in freely behaving rats running a linear track. This is a follow-up to their previous publication on the correlations between gamma activity in hippocampus in relation to cerebral blood flow measurements with the same instrument used here, but focused mostly on sleep (REM and NREM). Here, they performed a unique set of experiments using this powerful tool of doppler imaging developed by this group which offers the capacity to record blood flow with high temporal precision and good spatial accuracy. This doppler-based technique is extremely powerful and has several advantages compared to fMRI including much higher temporal resolution and the ability to combine this method with other techniques (here electrophysiology) in freely behaving rats. Certainly, this technique has a bright future.

Particularly, this study investigates the changes in CBV and theta and the gamma frequencies in association with speed in over trained rats running on a linear maze with two rewards on both ends. The recordings of the doppler responses were done coronally and in diagonal planes. The regions showing marked changes in CBV were the parietal cortex, retrosplenial, all of hippocampus regions and dorsal thalamus. Interestingly, the time lag of the CBV in association with the peak speed showed that the thalamus showed a short lag compared to the hippocampus and cortex, with the dorsal thalamus and dentate gyrus showing the shortest time lags. There appears to be a sequence in BV when cross-correlated to speed. Similarly, to their previous study done in sleep, there appears to be a high correlation between theta and high-frequency gamma, and a weak correlation with low-gamma and CBV during running. Some of the most interesting results show that the dorsal hippocampus and the thalamus have increased CBV whereas neocortex tend to show an inverse relation to CBV in relation to continuous running episodes suggesting that these regions could be perfused by different vascular systems. The strength of this study is to use a combination of CBV measurements and field recordings in a stereotyped linear running task. Several results are novel and interesting, and this study represents a novel characterization of CBV changes recorded simultaneously in different brain areas. However, the changes observed are difficult to interpret and the significance of the results difficult to put into broader context. This study is definitively new and important but I find it hard to understand the implications, but admittedly this is due in part to the fact that there are almost no studies that have investigated this phenomenon in as much detail and moreover in freely behaving animals. Because the results are novel but the interpretation difficult, the authors may wish to clear-up some of the result sections.

We thank the reviewer for these positive comments. We agree on the fact that the interpretation of our results is complicated. We have therefore modified the Discussion section in depth and made our best efforts to ease the interpretation of results by clearing-up the Results section and Figures 3, 4 and 5.

Comments:

-While the results shown in Figs 1 and 2 are noteworthy, the sequential activation displayed in Fig. 3 is less clear. While overall patterns within regions (ie all hippocampus and all thalamus) appear convincing, the sequential activation (Fig. 3C and D) and its smaller regional distribution of CBV is less convincing. While some sequential distribution appears significant (Fig 3C, left lane) for hippocampus is clear, that on the right lane is not and does not show the same sequence of CBV? Furthermore, the results shown in C don't seem to fit the results for the group distribution in d. However, the distribution of coronal and sagittal results appears comparable. Is there a real distribution of the tri-synaptic loop in hippocampus given the data in d?

We agree that the sequential activation displayed in Fig. 3C and 3D requires clarification. First and foremost, the delays between speed and CBV activation were computed with a precision of 200 milliseconds only, which is suitable for the large regional distributions but too coarse to render the small delays between hippocampal sub-regions. Additionally, we realized that the legend in Fig. 3C right lane contained an error: DG was mislabeled CA1 and CA1 mislabeled DG.

In order to precisely measure the timing of regional and sub-regional CBV profiles, we have re-processed the Doppler movies to increase the sampling rate of the CBV signal to 100 Hz. This is possible because the raw ultrasound data (IQ matrices) is acquired continuously at a pulse repetition frequency of 500 Hz. Thus, using sliding windows of 200 milliseconds (with a 190 millisecond overlap) we were able to generate high-definition movies containing one Doppler frame every 10 milliseconds, each frame being calculated on 200 milliseconds of raw data. This allows for a better measure of the delays between running speed and hemodynamics activations. We found the new group distributions more consistent than the previous ones, both in terms of consistency with individual distributions and between coronal and diagonal groups. This new analysis is shown in Fig. 3D. For the sake of clarity, we have removed the low-correlating regions (Determination coefficient $R_{max}^2 < .2$, i.e. Correlation coefficient $R_{max} < .448$). The information for other regions is available in Supplementary Fig. 5.

To settle the question of the distribution of the tri-synaptic circuit, we have computed the delay differences between the strongly-correlating regions (namely dorsal thalamus, retrosplenial cortex and hippocampal subfields) and performed statistical analysis. A clear sequence emerges in both recording groups dThal/RS cortex \rightarrow Dentate Gyrus \rightarrow CA1/CA3 regions. We found a significant delay difference between dorsal thalamus/dentate gyrus and between dentate gyrus/CA1 region. On the diagonal planes the CA3 region labeling was more challenging, probably explaining the small discrepancies in absolute timing. Overall, the sequence was conserved and clearly visible in individual recordings. It might not be sufficient to call this sequence ‘trisynaptic activation’, but this analysis strongly supports the idea of independent perfusion networks within the hippocampus and is coherent with the concept of trisynaptic network. Results from panel E were merged with those from panel D to present the vascular propagation in a more synthetic fashion.

-As for the electrophysiological recordings using tungsten wires, it was not indicated clearly where the wires were located (histology?). Obviously, recordings of theta and gamma, as well as their phase-locking values will depend on which layers was recorded in hippocampus. Therefore, it is unclear how reliable are the cross-correlation of theta with faster frequency oscillations. Moreover, even if this coupling was measured accurately (using linear probes for example), the interpretation of the relationship between the coupling and **the CBV** would likely be difficult to explain. What would be the interpretation?

We have provided details about electrode implantation in the Methods sections and pictures of histology after the lesioning protocols at recordings sites in Supplementary Figure S1. The actual design is based on handmade electrodes with minimal spacing of 500 microns between recording sites and a maximal number of 6 electrodes per bundles. This allows to observe the characteristic phase reversal between the superficial and deep layers of the dorsal hippocampus (Bragin et al. 1995), but not to quantify the cross-frequency coupling as with linear probes. See Figure S1 for further details.

If theta-gamma coupling in the hippocampus could be measured accurately in this experimental condition, we could investigate whether the coupling strength (as measured by a modulation index) explains the modulation of brain hemodynamics more than the power of fast gamma oscillations per se. Indeed, increased power of gamma oscillations does not necessarily entails stronger phase-frequency coupling and conversely. If this were the case, a possible interpretation would be that increased cross-frequency coupling means more efficient activation of distributed cellular assemblies, possibly triggering long-term potentiation processes that could in turn require more energy. We have added a paragraph about this point in the Discussion section.

-Fig 4 shows the relationship between the CBV and the field recordings. The high correlation between theta and high-gamma, but not low-gamma, are expected since previous studies have shown the correlation between these frequencies and running speed per se. Also, the data showing that the field recordings preceded the CBV change was also noted in their previous study.

We agree that these results are of less importance than the other findings presented here. However, it is interesting to note that while fast gamma oscillations correlated with CBV stronger than theta oscillations during REM sleep, it is the opposite during wake. To provide a full picture of the neurovascular interactions during wake, we have computed both local and distant LFP-CBV correlations with electrodes located in the hippocampus and motor cortex (new sets of experiments). We found a very clear decoupling of CBV signals in the primary motor cortex both to hippocampal theta/gamma oscillations and to local cortical oscillations. This nicely aligns with previous studies showing that neurovascular coupling is region-dependent (Devonshire et al. 2012) and that CBV in frontal brain regions is decoupled from LFP during locomotion in head-restrained mice (Huo et al. 2014). These results are presented in Fig. 4, which was simplified.

-The results shown of Fig 5 and 6 showing the differential dynamics of the CBV changes in different areas in relation to trial numbers is very interesting and unexpected. However, the interpretation is more difficult. The hemodynamic reshaping probably does not reflect changes in the tuning of place fields or plasticity-related shifts in the place cells as cited by the authors since place cells are formed rapidly when running in the same linear maze and stabilize rapidly (backward shifts are reduced after a dozen trials) after a dozen of running trials. Therefore, the timeframe of the increase in CBV in hippocampus observed here and place cells changes do not occur in the same timescale. Anyhow, this issue could be settled by using an NMDA antagonist which are known to block these backward shift changes.

We thank the reviewer for these positive comments. We agree that the plasticity-related place cells shifts occurs on a different timescale and is thus most probably unrelated with the vascular potentiation observed in the hippocampus on the timescale of minutes. We have thus modified the corresponding sentences in the Discussion section in light of these comments.

-How can the CA2 area be accurately and correctly measured since it's a fairly small structure and the area measured by the system appears at the limit?

We have registered two different atlases (Paxinos Atlas and Waxholm atlas) on our ultrasound acquisitions to derive regions of interest. The second atlas is based on MRI acquisition and provides labeling of small anatomical areas such as CA2, which we used. However, mismatches between the true location of small regions and the regional parcellation are likely. Thus, to avoid inaccurate labeling of these small structures we have simply removed CA2 (and structures below 50 pixels) from our analyses.

Reviewer #2 (Remarks to the Author):

Bergel et al. applied a functional ultrasound imaging method to map the hemodynamic responses from a single brain slice (two orientations) with concurrent electrophysiology in the hippocampus of normal behaving rats during running. The overall technological establishment is impressive. However, the experimental design needs to be clarified and the interpretation of results needs to be better justified with good control studies. Also, in the discussion section, these authors covered several key observations without a thorough discussion of the literature, leaving the conclusion less convincing. Several major concerns should be dealt with to improve the overall quality of this manuscript before it can be published.

We thank the reviewer for this positive comment and the different suggestions to improve the quality of the paper. We have revised the paper thoroughly to clarify the experimental design and have modified the Discussion section in depth to provide a thorough linkage to the existing literature as advised. Importantly, we have performed a new set of control studies on 3 animals: two animals were implanted with LFP electrodes in the dorsal hippocampus and motor cortex and imaged over a coronal section of the brain (Bregma +3.00 mm) encompassing primary and secondary motor cortices, other frontal cortices (prelimbic, anterior cingulate, infralimbic) and striatum and one animal was imaged on the same plane, but without electrodes. This new dataset thus includes 12 recordings on these 3 animals (8 of

which with LFP recordings). The results have been integrated to Figure 2, 4 and 6 and are detailed in a point-by-point fashion below.

1. This work is done from the pioneering group on fUS led by Dr. **Tanter**. Despite the great track record of the application of fUS to map brain dynamics, it remains hard for the readers to understand what is exactly the signal detected by fUS. The authors claimed the CBV signal measurement with some references. But, the cited references do not show a clear definition or explanation of how CBV signal was measured with fUS. If the functional map is CBV % change. How was it calculated? It should be clearly explained in the method section. It is crucial to specify the signal detected by fUS, which is also critical for the interpretation of the lag times in comparison to the existing fMRI studies. In particular, the authors need to discuss their work in the context of the existing literature of hemodynamic mapping and neurovascular coupling in animals.

We have updated the Methods section to provide further details on how CBV maps were built from the underlying ultrasound echoes and how pixel CBV % change was calculated (Section CBV maps & Baseline Averaging).

On a mathematical point of view, the fUS signal in a voxel arises from ultrasound echoes generated by echogenic particles (acoustic scatterers) moving at different speeds along different orientations during a typical time window of 200 milliseconds (the duration of a rat's cardiac cycle) (Macé et al. 2011, Macé et al. 2013). After Singular Value Decomposition (SVD) filtering, we are able to reject global tissue motion (due to pulsatility) and keep only echoes generated by red blood cells. In practice, the fUS signal is influenced by multiple factors: number and size of vessels contained in a voxel, vessel orientation, scatterer velocity and variations in vessel diameter. These parameters are not accessible and influence each other, meaning that absolute values of CBV or CBF are difficult to estimate.

Estimations of CBF and CBV changes relative to a baseline period are however possible. Indeed, once a baseline image has been acquired, the number, size and orientation of vessels remain constant. Upon local vasodilation or constriction, only RBC speed and changes in vessel diameter influence the fUS signal. These two parameters (RBC speed and diameter change) affect the Doppler spectrum differently: 1 - Variations in RBC velocity shift the mean value of the Doppler spectrum (leaving its global power unaffected), a parameter used to build Color Doppler images. 2- Variations in vessel diameter change the number of scatterers inside a voxel and directly increase or decrease the full energy of the Doppler spectrum, a parameter used to build Power Doppler images. Importantly, this is the case because most vessels are smaller than the size of a voxel, otherwise vessel diameter changes would affect several voxels, but not a single one. Ultimately, Power Doppler images reflect relative CBV % changes while Color Doppler images reflect relative CBF % changes.

Concurrent recordings of neural activity, individual blood vessel dynamics and functional ultrasound signal *in vivo* are now available to settle this point. Our group has been collaborating with Dr. Charpak's group in Paris and we have recently published a new study entitled *Transfer functions linking neural calcium to single voxel functional ultrasound signal* in Nature Communications. This study shows that fUS signal can be predicted from calcium recordings and single vessel hemodynamic through robust transfer functions in a wide set of stimulation paradigms, thus establishing the neural and vascular underpinnings of the fUS signal. The qualitative temporal evolution of the fUS signal at the mesoscopic level in a single pixel is found comparable to the red blood cells speed estimated in the same pixel at the microscopic level using 2-photon microscopy.

This complements another study published in 2019 by the same group entitled *Microscopic and mesoscopic imaging of sensory responses in the same animal*, again in Nature Communications, where they acquired fMRI signal, fUS signal and single vessel dynamics in the same animal in response to different odors. In particular, they found short time lags (on the order of hundreds of milliseconds to a second) for vascular responses measured by TPLSM and fUS (Fig. 4), whereas BOLD responses were found to be slower and peaked later on the order of tens of seconds (Fig. 6). These two studies, though

limited to the olfactory bulb of anesthetized animals, clearly establish the transfer functions between neural activity and single vessel hemodynamics and the fUS signal. Additional work is required to establish the same transfer functions in other structures, ideally in awake animals.

a. The authors need to discuss the fMRI studies of the hippocampus with electrophysiology or calcium recordings in non-human primates and rodents. The specific hemodynamic responses have been well described in terms of lag times or correlation features. The very short duration and lag time of the “CBV” signal reported in this manuscript needs better justification.

We have modified the Discussion in depth to include the fMRI studies of the hippocampus together with electrophysiology and behavior. In particular, we have compared the timing of the hemodynamic responses in blood-oxygen-level-dependent (BOLD) studies to the ones obtained by other groups and techniques in particular using intrinsic optical imaging (IOS), two-photon laser-scanning microscopy (2PLSM) and functional ultrasound (fUS). Importantly, lag times between fUS and electrophysiological signals (Bergel et al. 2018) and between fUS and behavioral cues (Dizeux et al. 2019) have been precisely measured and relate tightly to timings measured using optical techniques (Pisauro et al. 2013, Huo et al. 2014, Ma. et al. 2016, Mateo et al. 2017).

We reconcile the apparent discrepancies in terms of lag times (shorter using IOS, 2PLSM, fUS than the ones found in BOLD studies) arguing that these techniques do not measure the same parameters and therefore do not have access to the same temporal dynamics: BOLD imaging measure a ratio between deoxyhemoglobin (HbR) and oxyhemoglobin (Hillman et al. 2014), a parameter that has much slower dynamics, than cerebral blood volume (CBV), which dynamics are faster and can be a confounding factor in BOLD studies. Again, this has been recently established in terms of calcium recordings, TPLSM, fUS and fMRI in the same animal by Dr Charpak’s group.

b. The authors also attributed some of their unique observations to astrocyte function involved in neurovascular coupling, but a better job should have been done to elaborate on how astrocyte function might be a potential cause. During locomotion, there are decent studies to record the astrocyte function in the cerebellum and the cortex in normal behaving mice. Please provide a thorough discussion. Importantly, A series of studies show the global astrocyte activation upon locomotion or arousal state changes. Then, how will astrocyte function be coupled to hemodynamic responses in the brain during locomotion? The authors need to provide better linkage between their observation and the existing literature.

This is also a critical point and we thank the reviewer for raising it. Astrocytes are involved in a wide range of functions but can directly trigger local vasodilation or constriction through the release of vasoactive agents (Attwell et al. 2010, Cauli et al. 2011, Iadecola 2015). The study by Paukert and collaborators published in *Neuron* in 2014 established that astrocytes are globally activated during locomotion. Investigating the coupling between astrocytic activation and hemodynamic responses during locomotion is an open and important question. Based on the existing literature, we expect these two responses to be strongly coupled. Performing hemodynamic measurements together with astrocytic imaging in head-fixed animals during virtual reality locomotion test could settle this key question. We have updated the Discussion in order to provide a better link with these studies on astrocyte function during locomotion.

2. The LFP data acquisition. During locomotion, the LFP signal will be possibly contaminated with motion artifacts, the authors need to provide some detailed information on how their LFP signal is avoided from the artifacts or how they process the LFP data. In particular, if no robust fUS signals were reliably observed in some part of cortices. What will the LFP signal look like? It should be used as negative control data to justify the LFP signal detected in the hippocampus. It is a critical experiment to justify the existing results.

We have updated the Methods section to provide further details on LFP artifact rejection and electrode implantation, together with pictures of histology after the lesioning protocols at recordings sites (Supplementary Figure S1) and with details about the LFP recordings in motor cortex and hippocampus during locomotion (Supplementary Figure S5).

In order to provide a negative control to the LFP signal detected in the hippocampus, we have analyzed the LFP-CBV coupling at other electrode recording sites (LFP electrodes in the parietal cortex, located at the top of the bundle implanted above the dorsal hippocampus, See Fig. S1) and on the new dataset acquired for the revision of this paper (animals implanted in the motor cortex and dorsal hippocampus). Overall, we found that the LFP-CBV coupling is highly-region dependent. In particular, we found that local gamma oscillations (50-100 Hz) are associated with a reduction in the CBV signal in the primary motor cortex, while gamma oscillations are associated with an increase in CBV in the hippocampus (see Figure 4).

This aligns nicely with previous results in the literature showing that motor cortex is often suppressed during repetitive behavior (Ebbesen & Brecht, 2017) and that there is a decoupling between neural and vascular signal in the motor cortex of head-fixed running mice (Huo et al. 2014). One possible underlying mechanism is that LFP partly emerges from inhibitory transmission that depresses overall network activity. So, the absence (or presence) of robust LFP signal cannot be equivocally associated with a reduction (or increase) in CBV signal but shall be studied in every region independently. We have performed such an analysis for 4 different brain structures in Fig. 4.

3. Some positive controls are missing to strengthen the arguments of the work in hippocampus. The authors present some highly variable fUS signals in different brain regions with only LFP recording in the hippocampus (also across trials). Thus, it is crucial to first identify the activation in the motor cortex or cerebellum during locomotion as the starting point to validate their method. Also, the LFP recording of the cortex is a crucial control signal to be presented when interpreting the fUS signal in the cortex during locomotion (refer to point 2).

We agree with the reviewer and performed new experiments. In order to provide a positive control to the work in the hippocampus, we have recorded the CBV activation in frontal cortices and striatum in 3 new animals (recording plane Bregma = +3.00 mm), two of which were implanted with electrodes in the dorsal hippocampus of one hemisphere and in the motor cortex in the other hemisphere. The main finding of this control study is that the primary motor cortex is deactivated during locomotion (20 % reduction) and activated (20 % increase) a few seconds after, corresponding to the time period when the animal gets its reward through the water port (see fig 2a right panel). Nicely, this is well in line with the literature showing that the primary motor cortex is deactivated when a rodent engages in repetitive locomotion behavior but correlates with task engagement and complex facial or body movement, which is the case when the animal takes up the reward. These results are presented in Fig. 2.

Additionally, like other cortical structures motor cortex activations were found to decrease over time showing an inverse pattern as the one observed in the hippocampus, while fast gamma oscillations were found relatively stable. We thank the reviewer for advising us to perform these important supplementary acquisitions. This strengthens again the argument that the work in the hippocampus is region-specific and accompanied by fast gamma oscillations locally (see Fig. 6 and 7). In particular, we demonstrate that the inter-trial coupling between CBV signals and LFP high-frequency oscillations was specific to electrodes located in the hippocampus: indeed, the same analysis was performed for electrodes in the hippocampus (CA1 region, site 2 Fig. S1) and in the parietal cortex (RS cortex, site 1 Fig. S1) and found that high correlations were only found for hippocampal electrodes. We also performed statistical analysis to assess the significance of these correlations. These new analyses are presented in Figure 7.

4. Statistics need to be better clarified. For example, in Fig 6 and 7, how many rats were used to make the overall presentation? Are those observations reproducible?

We have provided further details about the number of rats and recording sessions per rat in Fig. 6 and 7. We also performed statistical analyses to assess the significance of the potentiation/depression patterns observed in the CBV maps, and how they relate to LFP signal. These observations were fairly reproducible across individuals as shown by the individual data points shown in Fig. 6A and 6B.

5. The statement of vascular potentiation or plasticity is missing justification. First, based on the method description. The rats were anesthetized for 20-25min and wait for 40min before testing with 30 min duration. How can they be sure that the time-dependent responses are not due to the washout of the anesthetic effect after the rats started “running”? Secondly, do they perform another consecutive round of training from the same animals with varied resting time and test if the increased hippocampal fUS signal remains? Thirdly, a positive control site, e.g. motor cortex or cerebellum, is needed to examine if there is a real “vascular plasticity” by comparing it with the concurrent LFP (increased vascular responses do not mean that there is vascular plasticity.)

We have performed fUS imaging during the recovery period from the isoflurane anesthesia, which showed a marked decrease in all brain regions when anesthesia was stopped. Animals usually recovered from anesthesia within minutes, but the fUS signal needed an additional 5 to 10 minutes to go back to baseline depending on the brain region (see Figure S9). We let the animal recover for another 30 minutes and checked visually that the fUS, electrophysiological and behavior patterns were stable and resembled those observed during wake. This is a first argument to rule out a possible implication of the anesthesia in the modulation of hemodynamic patterns observed.

Another argument follows from the fact that though anesthesia duration was fairly comparable across sessions (15 to 20 minutes), the actual start of the running session was quite variable across recordings and showed no clear correlation with the potentiation patterns observed in the hippocampus. It was complicated to perform another round of training in the current setup because we needed to record for 30 to 40 minutes to gather a sufficient number of imaged runs (only 1 every third run could be imaged in this setup) per session, a time after which the animal was satiated. Reducing water reward size or introducing a delay between the locomotion and the reward could help reach satiation later and would thus allow for two consecutive running sessions interleaved with rest periods. From our perspective, it is unlikely that such delay would affect vascular potentiation, though this remains to be checked experimentally.

As for the last point, we agree that increased vascular response does not necessarily mean vascular plasticity. To us, the term “plasticity” was the best denomination we found to summarize the adaptive modulation or reshaping of vascular patterns observed here. The fact that, at least in the hippocampus, CBV changes were mirrored by increased power in high-frequency oscillations, while these remained stable at the cortical sites supports the idea of local vascular plasticity in the hippocampus. Yet, to prevent potential misunderstanding, we propose to replace vascular plasticity with vascular modulation.

Overall, this work shows some promising and interesting aspects of fUS brain mapping of normal behaving animals. The technique is interesting and inspiring, but the scientific arguments of this manuscript need to be improved thoroughly.

We thank the reviewers for these positive comments and hope to have addressed their concerns.

REVIEWERS' COMMENTS

Reviewer #1 (Remarks to the Author):

Reviewer 1: The results of this manuscript are truly original. It shows how brain oscillations are associated with cerebral blood flow and they show how CBV is different from region to region with the largest differences taking place in the hippocampus and motor cortex when the animal performs a very stereotyped running behavior in a linear maze with rewards. The authors have answered most of my queries and have made the necessary changes to the manuscript. Besides minor questions (see at the end), which the author may decide to make very minor additions to the text (or to a figure), the results of this manuscript deserve to be published. This publication will likely raise a lot of new questions in the field.

Authors previous answers: We thank the reviewer for these positive comments. We agree on the fact that the interpretation of our results is complicated. We have therefore modified the Discussion section in depth and made our best efforts to ease the interpretation of results by clearing-up the Results section and Figures 3, 4 and 5.

Previous reviewer 1 comments:

-While the results shown in Figs 1 and 2 are noteworthy, the sequential activation displayed in Fig. 3 is less clear. While overall patterns within regions (ie all hippocampus and all thalamus) appear convincing, the sequential activation (Fig. 3C and D) and its smaller regional distribution of CBV is less convincing. While some sequential distribution appears significant (Fig 3C, left lane) for hippocampus is clear, that on the right lane is not and does not show the same sequence of CBV? Furthermore, the results shown in C don't seem to fit the results for the group distribution in d. However, the distribution of coronal and sagittal results appears comparable. Is there a real distribution of the tri-synaptic loop in hippocampus given the data in d?

Previous author comments: We agree that the sequential activation displayed in Fig. 3C and 3D requires clarification. First and foremost, the delays between speed and CBV activation were computed with a precision of 200 milliseconds only, which is suitable for the large regional distributions but too coarse to render the small delays between hippocampal sub-regions. Additionally, we realized that the legend in Fig. 3C right lane contained an error: DG was mislabeled CA1 and CA1 mislabeled DG.

-2nd review: Thank you for the corrections and clarification.

In order to precisely measure the timing of regional and sub-regional CBV profiles, we have re-processed the Doppler movies to increase the sampling rate of the CBV signal to 100 Hz. This is possible because the raw ultrasound data (IQ matrices) is acquired continuously at a pulse repetition frequency of 500 Hz. Thus, using sliding windows of 200 milliseconds (with a 190 millisecond overlap) we were able to generate high-definition movies containing one Doppler frame every 10 milliseconds, each frame being calculated on 200 milliseconds of raw data. This allows for a better measure of the delays between running speed and hemodynamics activations. We found the new group distributions more consistent than the previous ones, both in terms of consistency with individual distributions and between coronal and diagonal groups. This new analysis is shown in Fig. 3D. For the sake of clarity, we have removed the low-correlating regions (Determination coefficient $R_{max} < .2$, i.e. Correlation coefficient $R_{max} < .448$). The information for other regions is available in Supplementary Fig. 5.

1st reviewer: Yes that really helps, the activation sequence is clear.

Previous comments from Authors: To settle the question of the distribution of the tri-synaptic circuit, we have computed the delay differences between the strongly-correlating regions (namely dorsal thalamus, retrosplenial cortex and hippocampal subfields) and performed statistical analysis. A clear sequence emerges in both recording groups dThal/RS cortex -> Dentate Gyrus -> CA1/CA3 regions. We found a significant delay difference between dorsal thalamus/dentate gyrus

and between dentate gyrus/CA1 region. On the diagonal planes the CA3 region labeling was more challenging, probably explaining the small discrepancies in absolute timing. Overall, the sequence was conserved and clearly visible in individual recordings. It might not be sufficient to call this sequence 'trisynaptic activation', but this analysis strongly supports the idea of independent perfusion networks within the hippocampus and is coherent with the concept of trisynaptic network. Results from panel E were merged with those from panel D to present the vascular propagation in a more synthetic fashion.

-2nd reviewer comment: That is clear.

Previous comments from Authors: -As for the electrophysiological recordings using tungsten wires, it was not indicated clearly where the wires were located (histology?). Obviously, recordings of theta and gamma, as well as their phase- locking values will depend on which layers was recorded in hippocampus. Therefore, it is unclear how reliable are the cross-correlation of theta with faster frequency oscillations. Moreover, even if this coupling was measured accurately (using linear probes for example), the interpretation of the relationship between the coupling and the CBV would likely be difficult to explain. What would be the interpretation?

We have provided details about electrode implantation in the Methods sections and pictures of histology after the lesioning protocols at recordings sites in Supplementary Figure S1. The actual design is based on handmade electrodes with minimal spacing of 500 microns between recording sites and a maximal number of 6 electrodes per bundles. This allows to observe the characteristic phase reversal between the superficial and deep layers of the dorsal hippocampus (Bragin et al. 1995), but not to quantify the cross-frequency coupling as with linear probes. See Figure S1 for further details.

If theta-gamma coupling in the hippocampus could be measured accurately in this experimental condition, we could investigate whether the coupling strength (as measured by a modulation index) explains the modulation of brain hemodynamics more than the power of fast gamma oscillations per se. Indeed, increased power of gamma oscillations does not necessarily entails stronger phase-frequency coupling and conversely. If this were the case, a possible interpretation would be that increased cross-frequency coupling means more efficient activation of distributed cellular assemblies, possibly triggering long-term potentiation processes that could in turn require more energy. We have added a paragraph about this point in the Discussion section.

-1st reviewer comments: Thank you for the explanation for the histology of the electrode placement. Though, the mechanisms and the role of cross-frequency coupling is still unclear and has this could have an effect on CBV. One leading explanation might be the increased interaction between fast-spiking interneurons and principal cells with changes in synaptic plasticity. It remains to be determined how theta and faster-gamma is coupled mechanistically, they are likely to be mediated by small changes in the coupling of processes, whether by another group of neurons, or local excitatory and inhibitory interactions that may not necessitate significant changes in energy demand (blood flow).

Previous reviewer 1 comments: -Fig 4 shows the relationship between the CBV and the field recordings. The high correlation between theta and high-gamma, but not low-gamma, are expected since previous studies have shown the correlation between these frequencies and running speed per se. Also, the data showing that the field recordings preceded the CBV change was also noted in their previous study.

Previous author comments: We agree that these results are of less importance than the other findings presented here. However, it is interesting to note that while fast gamma oscillations correlated with CBV stronger than theta oscillations during REM sleep, it is the opposite during wake. To provide a full picture of the neurovascular interactions during wake, we have computed both local and distant LFP-CBV correlations with electrodes located in the hippocampus and motor cortex (new sets of experiments). We found a very clear decoupling of CBV signals in the primary

motor cortex both to hippocampal theta/gamma oscillations and to local cortical oscillations. This nicely aligns with previous studies showing that neurovascular coupling is region-dependent (Devonshire et al. 2012) and that CBV in frontal brain regions is decoupled from LFP during locomotion in head-restrained mice (Huo et al. 2014). These results are presented in Fig. 4, which was simplified.

Previous reviewer 1 comments: The results shown of Fig 5 and 6 showing the differential dynamics of the CBV changes in different areas in relation to trial numbers is very interesting and unexpected. However, the interpretation is more difficult. The hemodynamic reshaping probably does not reflect changes in the tuning of place fields or plasticity-related shifts in the place cells as cited by the authors since place cells are formed rapidly when running in the same linear maze and stabilize rapidly (backward shifts are reduced after a dozen trials) after a dozen of running trials. Therefore, the timeframe of the increase in CBV in hippocampus observed here and place cells changes do not occur in the same timescale. Anyhow, this issue could be settled by using an NMDA antagonist which are known to block these backward shift changes.

Previous author comment: We thank the reviewer for these positive comments. We agree that the plasticity-related place cells shifts occurs on a different timescale and is thus most probably unrelated with the vascular potentiation observed in the hippocampus on the timescale of minutes. We have thus modified the corresponding sentences in the Discussion section in light of these comments.

-Thank you

-How can the CA2 area be accurately and correctly measured since it's a fairly small structure and the area measured by the system appears at the limit?

We have registered two different atlases (Paxinos Atlas and Waxholm atlas) on our ultrasound acquisitions to derive regions of interest. The second atlas is based on MRI acquisition and provides labeling of small anatomical areas such as CA2, which we used. However, mismatches between the true location of small regions and the regional parcellation are likely. Thus, to avoid inaccurate labeling of these small structures we have simply removed CA2 (and structures below 50 pixels) from our analyses.

Reviewer 1 comments: thank you for the changes.

-1st reviewer additional new questions/comments:

While one main idea of the manuscript is that speed and/or acceleration-deceleration are computed by theta and higher frequencies gamma oscillations, some of the key brain regions for computing speed such as striatum and Entorhinal cortex are not analysed here. Is there some data on these areas, if so, it would be worth including these in the main text or in the supp figures, as these are of interests to researchers interested in the neural basis of navigation.

As to the remarkable lasting increase in CBV in hippocampus and a few other regions, the author may wish to mention that local control of vasculature may be region specific and that feedback regulation in the hippocampus may be wired in such a way to promote such a slow 'recovery' back to baseline which would favor supporting long-lasting plastic event. Its hard to imagine any plastic event occurring in such a stereotypical task (running in a linear maze), but the hippocampus, surprisingly, may be wired in this manner.

It appears that high-gamma and HFO are tightly coupled to CBV signal. Since this is somewhat of a surprise, I would at a minimum show electrophysiological traces of each of these events. When describing high-frequency oscillations, these must be sharp-wave ripples or ripples? Its unclear why this link with HFOs which usually occur when the animal is immobile (drinking, grooming, getting ready to go, or in slow-wave sleep). The authors must show example of these events and give a possible explanation of why they would be associated with CBV.

Reviewer #2 (Remarks to the Author):

The authors have answered all my concerns.

Reference:

The authors provide a clear explanation how the CBV was defined using fUS.

It will be better to cite the existing literature to show CBV measurement from individual arterioles with single-vessel fMRI and simultaneous neuronal calcium recordings.

Also, the author provides thorough discussion on the astrocytic function. It is also better to cite simultaneous fMRI and astrocytic calcium recording studies from literature.

Statistics:

The revised manuscript has provided sufficient analysis with different control studies.

Reviewer #1 (Remarks to the Author):

The results of this manuscript are truly original. It shows how brain oscillations are associated with cerebral blood flow and they show how CBV is different from region to region with the largest differences taking place in the hippocampus and motor cortex when the animal performs a very stereotyped running behavior in a linear maze with rewards. The authors have answered most of my queries and have made the necessary changes to the manuscript. Besides minor questions (see at the end), which the author may decide to make very minor additions to the text (or to a figure), the results of this manuscript deserve to be published. This publication will likely raise a lot of new questions in the field.

We thank the reviewer for these very positive comments and for their suggestions in increasing the quality of the paper. We have added the proposed additions to the main text.

-1st reviewer comments: Thank you for the explanation for the histology of the electrode placement. Though, the mechanisms and the role of cross-frequency coupling is still unclear and how this could have an effect on CBV. One leading explanation might be the increased interaction between fast-spiking interneurons and principal cells with changes in synaptic plasticity. It remains to be determined how theta and faster-gamma is coupled mechanistically, they are likely to be mediated by small changes in the coupling of processes, whether by another group of neurons, or local excitatory and inhibitory interactions that may not necessitate significant changes in energy demand (blood flow).

We thank the reviewer for these remarks. We have updated the Discussion section and modified the corresponding paragraph (page 18) to include these comments.

-1st reviewer additional new questions/comments: While one main idea of the manuscript is that speed and/or acceleration-deceleration are computed by theta and higher frequencies gamma oscillations, some of the key brain regions for computing speed such as striatum and Entorhinal cortex are not analyzed here. Is there some data on these areas, if so, it would be worth including these in the main text or in the supp figures, as these are of interests to researchers interested in the neural basis of navigation?

We thank the reviewer for raising this point. The entorhinal cortex's deep and distal location and vessel orientation (mainly parallel to the ultrasound transducer) make it difficult to image reliably in the current setup. A different probe placement and dedicated surgical design is probably needed to study this region. As for the striatum, the two coronal sections that were recorded here (Bregma +2.5 to 3.0 mm / Bregma - 3.5 mm to - 4.0 mm) only intersect the striatum (caudate putamen) at its very anterior and posterior ends. Some elements about the striatal activation during locomotion can be found over some recordings over the anterior coronal plane (Bregma +2.5 mm, Fig. 2C right bottom panel) showing an increase in CBV peaking 2.0 to 3.0 s after peak speed, and over the diagonal planes (Fig. 3C, right bottom line) showing that vascular activity shows a moderate but positive correlation with running speed peaking approximately 2.5 seconds after running speed. Again, the variability in atlas registration over diagonal planes makes this analysis complicated. Additional studies are required to investigate the dynamics of the whole locomotion network with functional ultrasound.

As to the remarkable lasting increase in CBV in hippocampus and a few other regions, the author may wish to mention that local control of vasculature may be region specific and that feedback regulation in the hippocampus may be wired in such a way to promote such a slow 'recovery' back to baseline which would favor supporting long-lasting plastic event. It is hard to imagine any plastic event occurring in such a stereotypical task (running in a linear maze), but the hippocampus, surprisingly, may be wired in this manner.

We thank the reviewer for these remarks. We have updated the Discussion section and modified the corresponding paragraph (page 20) to include these comments.

It appears that high-gamma and HFO are tightly coupled to CBV signal. Since this is somewhat of a surprise, I would at a minimum show electrophysiological traces of each of these events. When describing high-frequency oscillations, these must be sharp-wave ripples or ripples? It is unclear why this link with HFOs which usually occur when the animal is immobile (drinking, grooming, getting ready to go, or in slow-wave sleep). The authors must show example of these events and give a possible explanation of why they would be associated with CBV.

We thank the reviewer for raising this point. As mentioned in the introduction and based on previous studies, hippocampal gamma oscillations have been divided into three different subtypes, namely low gamma (30-50 Hz), mid gamma (50-100 Hz) and high gamma/epsilon/high-frequency oscillations (HFO) (100-150 Hz), all of which occur during locomotion. Importantly, high gamma/HFO strongly differ from ripple oscillations – which are observed when an animal is immobile (drinking, grooming, getting ready to move) or sleeps – in terms of amplitude, region of occurrence and activity time-course (Tort et al. 2013). These different gamma bands are clearly visible on a theta-phase time-frequency spectrogram.

To avoid confusion, we have added a sentence and a reference in the introduction about the difference between HFO and ripples (page 3) and slightly modified the titles of Figure 7C. We have also added traces of individual gamma events in Supplementary Figure 5, together with a time-frequency spectrogram showing the three distinct gamma sub-bands. Finally, we have added an explanation on why these different sub-bands may relate differently to the CBV signal in the Discussion section (page 18).

Reviewer #2 (Remarks to the Author):

The authors have answered all my concerns.

Reference:

The authors provide a clear explanation how the CBV was defined using fUS. It will be better to cite the existing literature to show CBV measurement from individual arterioles with single-vessel fMRI and simultaneous neuronal calcium recordings. Also, the author provides thorough discussion on the astrocytic function. It is also better to cite simultaneous fMRI and astrocytic calcium recording studies from literature.

Statistics:

The revised manuscript has provided sufficient analysis with different control studies.

We thank the reviewer for these comments and for their suggestions in increasing the quality of the paper.